# Targeted degradation of USP7 in solid cancer cells reveals distinct effects of deubiquitinase degraders and inhibitors

Nikolas Klink [1,2,13], Sebastian Urban[3,13], Johanna A. Seier[4,13], Bikash Adhikari [5], Martin P. Schwalm [6,7], Juliane Müller[5], Madeleine Dorsch[3], Philine Steinbach[4], Jennifer Jung [3,8], Markus Vogt [5], Farnusch Kaschani [9], Johannes Koch [10], Siska Führer [1,2], Markus Kaiser [11], Nina Schulze [10], Stefan Knapp [6,7,12], Elmar Wolf [5], Annette Paschen [4] ✉, Barbara M. Grüner [3,8] ✉ & Malte Gersch [1,2] ✉

Proteolysis-targeting chimeras (PROTACs) co-op the ubiquitin system for targeted protein degradation, creating opportunities to interrogate cellular functions of proteins through "chemical knockdown". However, matched pairs of protein degraders and inhibitors, that possess high specificity and chemical complementarity, for individual components of the ubiquitin system have remained scarce. This includes reagents to modulate activity and abundance of deubiquitinases (DUBs). Here, using an integrated chemical biology approach, we explore cellular functions of the DUB USP7 as a case study by comparing inhibition and degradation in melanoma and pancreatic cancer cells. Through the synthesis of a degrader library, we identify and characterize potent USP7 PROTACs for each cancer type. Proteomic and cellular analyses reveal that selective USP7 degradation modulates both shared and distinct protein sets across both cancers without affecting cell growth. In contrast, prolonged inhibitor treatment induces USP7-independent proteomic and metabolic dysregulation, highlighting important caveats for the cellular use of hydroxypiperidine-based USP7 inhibitors. Collectively, our work provides a comprehensively characterized chemical toolbox to distinguish on-target phenotypes which will aid the understanding of USP7 in malignant diseases. More broadly, our data emphasize the importance of increased specificity via PROTAC-mediated degradation and the potential of this modality to elucidate cell-line specific functions of DUBs.

Proteolysis-targeting chimeras (PROTACs) have emerged as powerful tools for both therapeutic applications and mechanistic biology. These hetero-bifunctional molecules enable rapid targeted protein degradation by hijacking the endogenous ubiquitin-proteasome system and have demonstrated remarkable potential in overcoming some of the limitations associated with conventional small molecule inhibitors[1,2]. Moreover, comparative studies assessing effects of targeted protein degradation vs. protein inhibition have revealed distinct outcomes facilitated by both modalities (e.g., for the transcriptional regulator BRD4)[3–5]

---

and enabled the discovery of non-catalytic protein functions (e.g., for the kinase Aurora A)[6,7]. However, such comparative analyses require matched pairs of well-characterized inhibitors and degraders, a requirement that remains largely unmet across most protein families.

The around 100 human deubiquitinating enzymes (DUBs) act as counterplayers of over 600 E3 ligases and represent a particularly compelling yet underexplored class of proteins where matched pairs of inhibitors and degraders are urgently needed[8,9]. These enzymes serve as critical regulators of cellular protein homeostasis and ubiquitin-mediated signaling. DUBs have emerged as promising therapeutic targets in cancer, neurodegeneration, and inflammatory disorders[10]. Despite their therapeutic potential, the development of selective DUB modulators has been challenging which has also hampered investigations into their cellular roles[11].

Ubiquitin-specific protease 7 (USP7) represents one of the most extensively studied human deubiquitinases[12]. USP7 has gathered significant attention as a regulator of key cellular processes including the p53-MDM2 axis (stabilizing both proteins depending on the cellular context), DNA damage response (DDR, stabilizing key DDR proteins, such as MDC1 or TRIP12), and transcription (through PRC1 complexes as well as RYBP)[13–16]. The multifaceted functions of this enzyme and its involvement in various cancer types have established USP7 as an attractive target for drug development, with several inhibitors being explored in preclinical studies[10,17–20].

Notably, while the targeted degradation of DUB proteins has been explored (including degraders for USP7)[21–26], studies describing matched pairs of well characterized inhibitors and PROTACs do not exist for DUBs. To address this unmet need, we selected USP7 as a case study and investigated inhibitor-PROTAC pairs in the biological context of pancreatic ductal adenocarcinoma (PDAC) and melanoma. USP7 is particularly suited to undergo a systematic comparative analysis of degradation vs. inhibition phenotypes for the following reasons:

(i). USP7 possesses numerous substrates and exhibits many context-dependent functions across different cell types[12,27–29]. This complexity demands well characterized and potent chemical tools to dissect its multifaceted biology.

(ii). In addition to catalytic activities, non-catalytic functions of DUBs are increasingly appreciated[30]. These include scaffold functions in chromatin regulation and protein complex assembly that are independent of deubiquitinase activity. A matched pair of USP7 inhibitor and degrader will thus provide valuable tools for dissecting these distinct functions.

(iii). USP7 is highly expressed in many cancers. However, its role in solid cancers remains poorly understood. This includes PDAC and melanoma wherein high USP7 levels are associated with poor prognosis[31–37].

(iv). While therapeutic interest has driven the development of many USP7 inhibitors, recent reports showcased that widely used compounds like P22077 lack sufficient specificity for USP7[11,38]. Although more specific USP7 inhibitors with different scaffolds as well as PROTACs targeting USP7 have been developed for application in leukemia cells[19,22,23,39], matched inhibitor-PROTAC pairs suitable for comparative studies in solid tumors have remained absent.

Here, we employ an integrated chemical biology approach to provide a comprehensively characterized matched pair of USP7 modulators, enabling systematic comparison of protein inhibition versus degradation (Fig. 1a). Our work establishes a framework for understanding DUB biology through complementary chemical perturbation approaches and provides validated tools for the systematic investigation of USP7.

## Results

### NK192 is a potent USP7 inhibitor with proteome-wide specificity and a functional exit vector

To develop a validated, specific and potent USP7 inhibitor and degrader pair, we first assembled a small collection of previously reported, structurally diverse USP7 inhibitors (Fig. 1b)[17–19,29,40]. Using the fluorogenic substrate Ubiquitin-Rhodamine110Gly (Ub-RhoG), we confirmed their inhibitory potency on recombinant, full-length USP7 (Fig. 1c). We focused on FT671 and Compound 5, which both feature a hydroxypiperidine core, display high inhibitory potency in vitro and are suitable for cellular use[17,19]. To facilitate the streamlined synthesis of USP7 degrader candidate molecules, we devised chimeric inhibitor NK192 which retains the 4-fluorophenyl-pyrazolo[3,4-*d*]pyrimidinone scaffold of FT671 and comprises (*R*)−3-phenylbutanoic acid as a fluorine-free equivalent of this motif in Compound 5. Chiral NK192 was readily prepared in four steps from commercially available starting materials (Supplementary Fig. 1a) and retained the high inhibitory potency on USP7 (Fig. 1c).

Both FT671 and Compound 5 are highly specific for USP7 when tested against related human USP family DUBs[17,19], yet whether they can potently bind non-DUB proteins had not yet been investigated. We therefore set out to confirm the target specificity of our chimeric USP7 inhibitor NK192 on a proteome-wide scale. Inspection of the crystal structure of USP7 in complex with FT671 revealed that the 4-fluorophenyl group pointed away from the catalytic domain[19] and suggested this site for chemical derivatization (Supplementary Fig. 1b). By replacing the fluorine with a piperazine linker, we next prepared the biotin-functionalized, NK192-based probe NK264 (Fig. 1b) which still potently engaged with USP7 (Fig. 1c). To comprehensively assess cellular targets of NK192, we conducted a competitive pulldown experiment. Lysates of Panc89 cells were treated either with DMSO or with a high concentration of NK192 (10 μM, 500x $IC_{50}$), followed by enrichment with NK264-functionalized streptavidin beads and mass spectrometric analysis. Our data revealed that NK192 exhibited very high specificity for USP7 (Fig. 1d, Supplementary Data 1) as USP7 was the only protein that was significantly competed away from the beads by NK192. These data established NK192 as a suitable, potent and specific ligand for the exploration of USP7 degraders to arrive at a chemically corresponding USP7 inhibitor and degrader pair. Moreover, the biotin probe yielded a functional exit vector in NK192 for the synthesis of bifunctional molecules.

### Identification of degraders for USP7 depletion in PDAC and melanoma

To arrive at a chemically diverse library of degrader candidate molecules, which would facilitate degradation of USP7 through induced proximity to the VHL E3 ligase[3], we employed a modular synthesis approach (Fig. 2a, steps A–D). Starting from a bromo-substituted NK192-equivalent as the USP7 ligand, we appended various mono-protected bifunctional amine-containing heterocycles via Buchwald-Hartwig[41] cross coupling reactions (step A). Following cleavage of the protecting groups, building blocks were either directly coupled to VHL ligand-linker conjugates (step B) or were further functionalized with linkers (step C), which were then fused to VHL ligand-linker conjugates (step D). After a first set of USP7 degrader candidates, which were designed with long linear linkers, did not yield promising degradation of USP7, we redirected our efforts toward a second-generation library of 15 compounds featuring mostly rigidified linkers (Fig. 2b). We aimed to limit flexibility between both ligands to enhance ternary complex formation and thereby degradation efficiency[42]. Synthesized compounds thus included a diverse set of hetero-, bi- and spiro-cyclic linkers, covering a wide variety of rigidity, geometry and linker length. Molecules comprising sterically demanding cubane and cyclohexyl groups directly adjacent to the VHL ligand building block were motivated by the presence of a hydrophobic pocket in the VHL E3 ligase

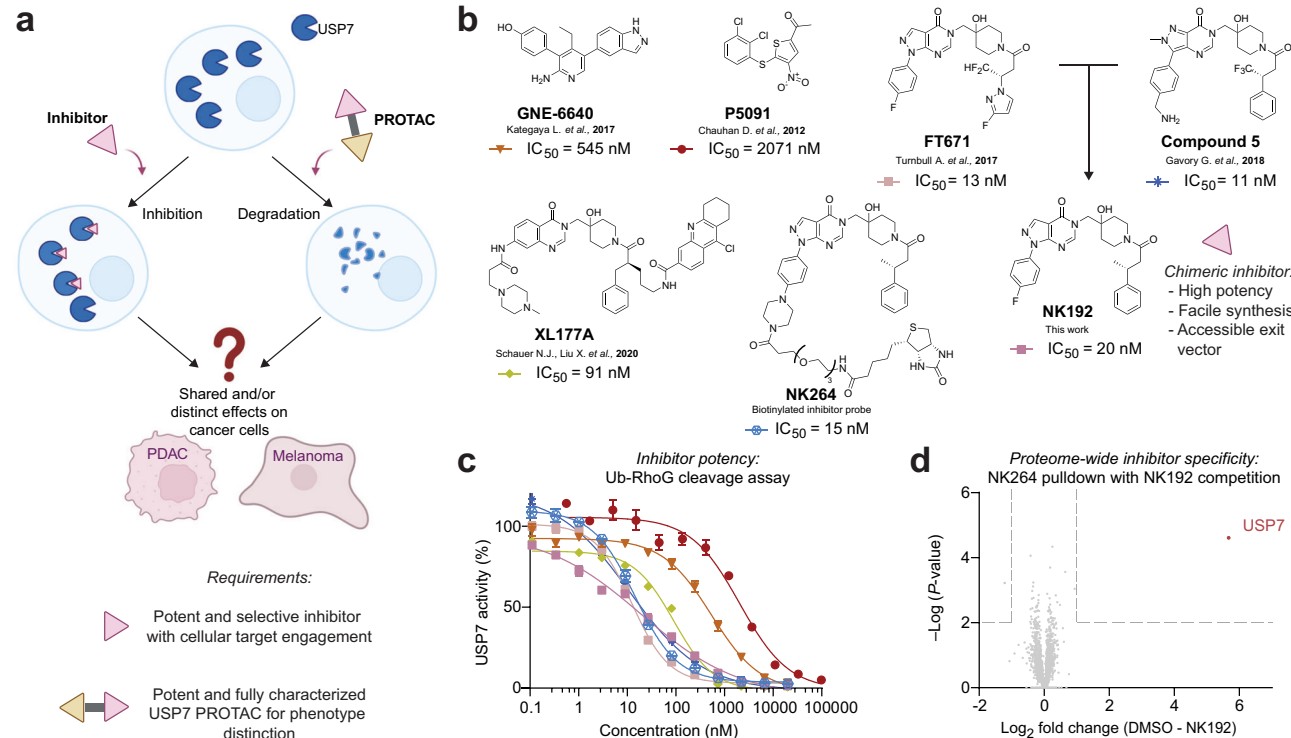

**Fig. 1 | Towards a comparative analysis of small molecule-mediated USP7 inhibition versus USP7 degradation phenotypes. a** Outline of this study. Comparative analysis of USP7 inhibition or degradation in PDAC and melanoma carried out using customized inhibitors and degraders. Partly created in BioRender. Klink, N. (2026) BioRender.com/f26vb8v. **b** Chemical structures of previously described USP7 inhibitors as well as of NK192 (chimera derived from potent USP7 inhibitors FT671 and Compound 5) which was used as USP7 ligand for the PROTAC library synthesis. Half-maximal inhibition ($IC_{50}$) values were derived from data shown in (**c**). NK264 represents a Biotin-functionalized variant of NK192, which was used for pulldown experiments shown in (**d**). **c** Ubiquitin-Rhodamine110Gly (Ub-RhoG) cleavage assay to determine inhibitory potency of compounds shown in (**b**). Full length human USP7 was incubated with the respective compounds, the fluorogenic substrate was added and residual activity was read out through fluorescence measurements. Data are shown as mean ± s.d. ($n = 3$ independent wells measured in parallel). **d** Competitive pulldown experiment to assess proteome-wide specificity of NK192. Biotin-functionalized NK264 was immobilized on beads. Proteins enriched from Panc89 lysate either treated with DMSO or with NK192 (10 μM, 500x $IC_{50}$) were quantified by mass spectrometry. *P*-values were determined from two-sided *t*-tests as implemented in Perseus.

---

which can be addressed by a cyclopropyl motif (e.g., in the VHL inhibitor VH298)[43]. Compounds were prepared with defined stereochemistry, purified by preparative reverse phase HPLC and subsequently used in cellular assays (see the Supplementary Information for chemical synthesis and compound characterization data).

To identify degraders for USP7 depletion in PDAC and melanoma, we treated both Panc89 PDAC and Ma-Mel-47 melanoma cells with compounds for 24 h and assessed USP7 protein levels by Western blot (Fig. 2c, d). We identified several PROTACs that effectively degraded USP7 in Panc89 cells, including NK225 and NK233 (Fig. 2e). While the degradation efficiency in Ma-Mel-47 was comparably lower, we also identified robust degradation of USP7 by NK225 and NK233. These compounds comprise trans-configured cyclohexane-1,4-dicarboxylic acid (NK225) or cubane-1,4-dicarboxylic acid (NK233) as well as a piperazine (NK225) or 4,4'-bipiperidine (NK233). Taken together, these results showed that USP7 PROTACs with different linker geometries facilitated degradation of USP7 with notable efficiency differences between both investigated cell lines.

## Characterization of two improved USP7 degraders customized to PDAC and melanoma cell lines

Encouraged by these results, we set out to further optimize these two degraders. To this end, we added a benzylic (*S*)-methyl group to the VHL ligand, which has previously been shown to improve affinity towards VHL[44]. This modification resulted in the third-generation PROTACs NK250 (derived from NK225) and NK266 (derived from NK233) (Fig. 3a). Additionally, we synthesized an NK225-based

negative control compound NK245 (Fig. 3a), which cannot bind VHL due to an inverted stereocenter at the hydroxyproline. Next, we treated both cell lines with these compounds and assessed USP7 levels by Western blot analysis (Fig. 3b, c). Both PROTACs NK250 and NK266, but not NK245, showed improved USP7 degradation in both cell lines. Consistently, cyclohexyl-PROTAC NK250 showed complete degradation in Panc89 cells, whereas cubane-containing PROTAC NK266 showed higher potency in Ma-Mel-47 cells.

To carry out an in-depth evaluation of both compounds, we assessed degradation potency in a concentration-dependent manner (Fig. 3d, e). Therefore, we treated both cell lines for 24 h with varying compound concentrations. We observed near complete degradation of USP7 with PROTAC NK250 in Panc89 cells near 100 nM (Fig. 3d, $D_{max}$: 98 %). A half-maximal degradation concentration value ($DC_{50}$) of 8 nM demonstrated excellent cellular potency (Fig. 3f). Likewise, NK266 showed a similarly strong reduction of USP7 levels in Ma-Mel-47 cells (Fig. 3e, $D_{max}$: 94 %), displaying a $DC_{50}$ value of 29 nM (Fig. 3g). Notably, NK266 also showed similar degradation efficiency as NK250 in Panc89 cells with a $DC_{50}$ of 49 nM and near complete degradation of USP7 (Supplementary Figs. 2a, b), whereas NK250 was not able to fully degrade USP7 at 1 μM concentration in Ma-Mel-47 cells after 24 h (Supplementary Figs. 2c, d). Based on these data, we decided on a concentration of 1 μM degrader for all following cell-based experiments and retained the use of NK250 for Panc89 and NK266 for Ma-Mel-47 cells.

To probe degradation kinetics, both cell lines were incubated with PROTACs, and USP7 levels were followed over 24 h (Fig. 3h, i).

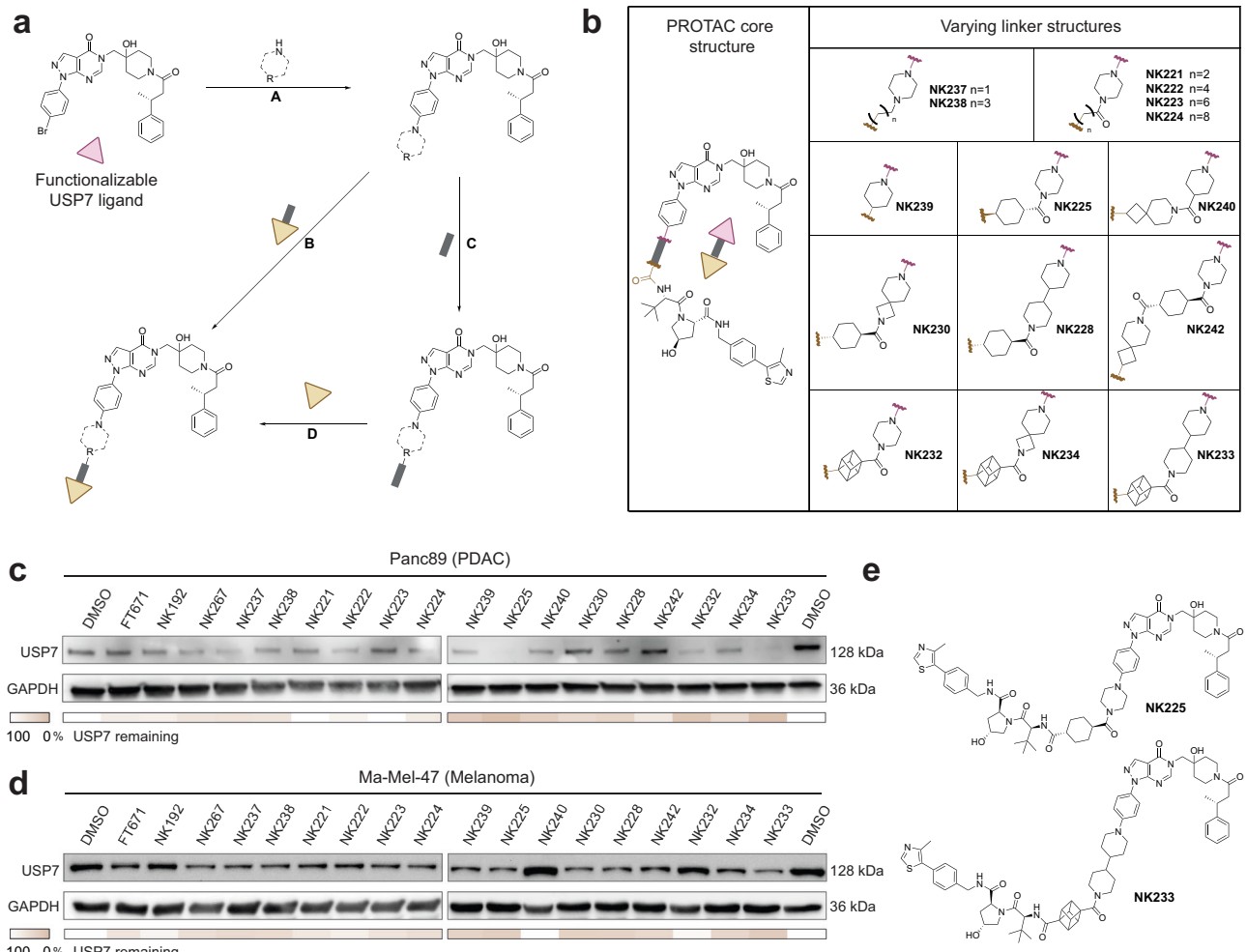

**Fig. 2 | Design and synthesis of USP7 targeting PROTACs. a** Schematic of the synthesis of USP7-targeting degraders. Functionalized USP7 ligands (top left, pink triangle) were either coupled directly to VHL-linker conjugates (yellow triangle with gray stick, through steps A and B) or first further functionalized with linkers followed by coupling to the VHL ligand (yellow triangle, through steps A, C and D). See the Supplementary Information for detailed chemical procedures. **b** Chemical structures of USP7-targeting degrader library. A general structure of the PROTACs is shown on the left, consisting of a USP7 binding ligand (top) and E3 ligase binding ligand (bottom) separated by a linker. Linker attachment points to both ligands are indicated by waved lines (black: USP7, red: VHL). Linkers are shown in the table on the right. **c**, **d** Assessment of USP7 levels upon treatment with compounds in Panc89 cells (**c**) and Ma-Mel-47 cells (**d**). Cells were treated with 5 μM of indicated compounds for 24 h, and cell lysates were analyzed through Western blots with indicated antibodies. Remaining USP7 levels were quantified by densitometry and are depicted as heatmaps. Uncropped blots are provided as Source Data. **e** Chemical structures of NK225 and NK233.

Treatment with NK250 resulted in rapid degradation of USP7 in Panc89 cells with protein depletion observed already after 6 h (Fig. 3h). In contrast, NK266 showed potent degradation only after 16 h in both Ma-Mel-47 and Panc89 cells (Fig. 3i, Supplementary Fig. 2e), in line with the weak effects of NK250 in Ma-Mel-47 cells (Supplementary Fig. 2f). *S*-Hydroxyproline PROTAC NK245 did not reduce protein levels in both cell lines during the same time frame, corroborating VHL-mediated USP7 depletion (Supplementary Figs. 2g, h). To confirm the mode of action of our degraders, we performed rescue experiments in which we blocked the degradation machinery involved in VHL-mediated protein depletion (Fig. 3j, k). Consistently, pretreatment of cells with either the NEDDylation inhibitor MLN4924 (which abrogates VHL E3 ligase activity), the proteasome inhibitor carfilzomib or a tenfold excess of VHL ligand NK249 (Supplementary Fig. 2i, to block VHL engagement) rescued PROTAC-mediated degradation of USP7. In addition, we confirmed USP7 degradation by immune fluorescence imaging in Panc89 cells (Fig. 3l). We detected a signal for USP7 predominantly localized to the nucleus, consistent with literature on USP7 localization[45], which was strongly reduced after treatment with NK250.

Together, these data demonstrated potent depletion of USP7 by the PROTACs NK250 and NK266 and confirm their mode of action as Cullin-Ring-E3 ligase (CRL)-dependent degraders.

## Quantitative assessments of USP7 degradation efficiency and PROTAC-mediated ternary complex formation

We next aimed to quantitatively investigate the mechanism driving their distinct degradation kinetics. We first established an additional orthogonal approach to measure cellular degradation efficiencies based on the HiBiT technology (Fig. 4a)[46]. To this end, we created an MV4-11 cell line which stably expresses N-terminally HiBiT-tagged USP7. The 11 amino acid HiBiT-tag can bind the LgBiT polypeptide, which results in an active luciferase, generating a signal proportional to the amount of intact HiBiT-USP7 protein (Fig. 4a). We then treated these MV4-11 cells with either NK250 or NK266 and observed dose-dependent reduction of USP7 levels after 6 and 24 h (Fig. 4b). In line with the results from Western blots, NK250 showed higher potency towards HiBiT-USP7 than NK266, particularly after 6 h. At this time point, NK250 showed a maximum degradation efficiency ($D_{max}$) of 87

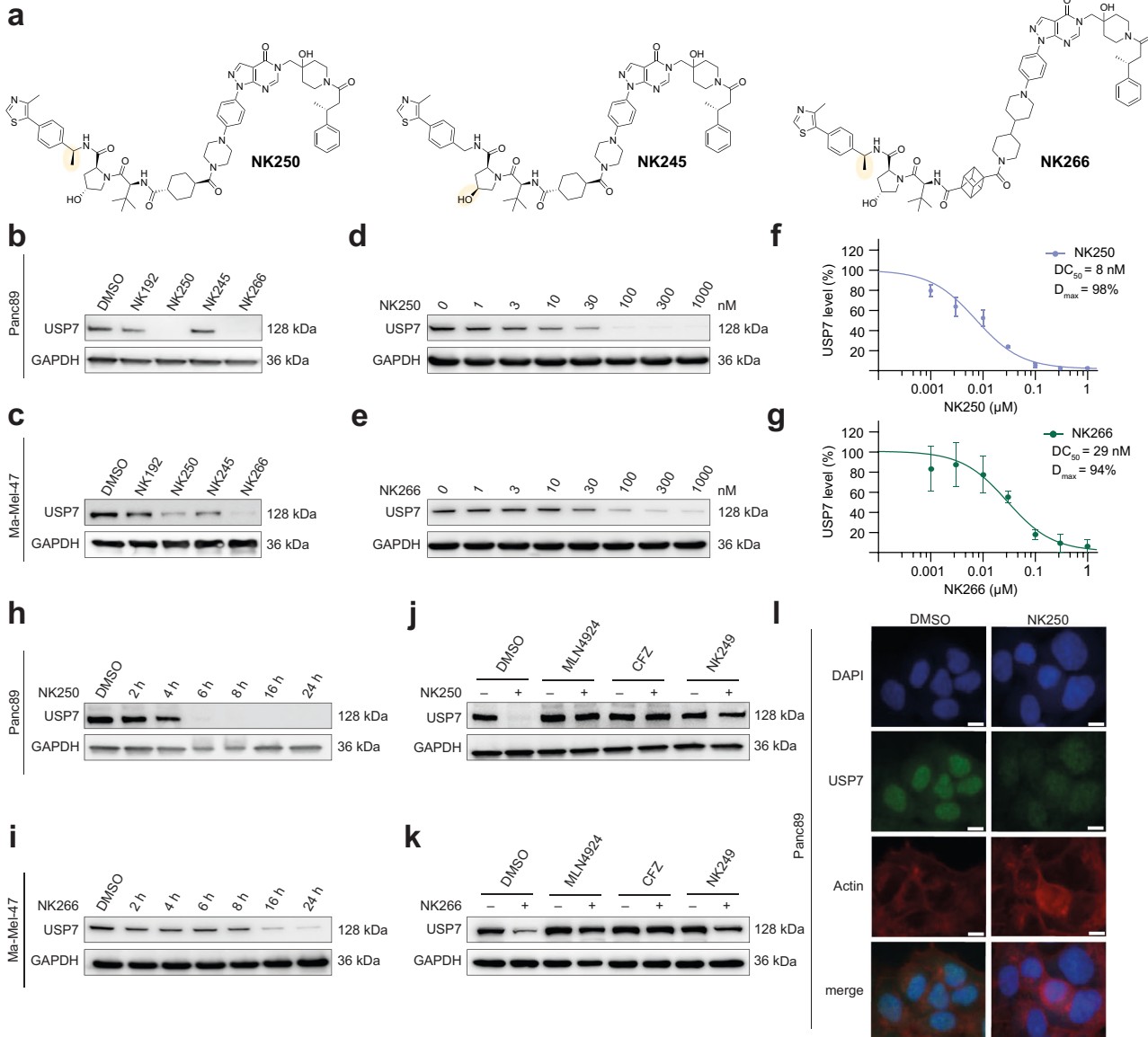

**Fig. 3 | Characterization of two potent USP7 degraders. a** Chemical structures of improved VHL-targeting PROTACs NK250, NK266 and non-VHL binding control compound NK245. The additional methyl groups in the VHL ligand in NK250 and NK266 (leading to enhanced VHL binding) as well as the inverted hydroxyproline stereocenter in NK245 (abrogating VHL binding) are highlighted. **b, c** USP7 degradation assays. Panc89 (**b**) and Ma-Mel-47 (**c**) cells were treated with 5 μM of indicated compounds for 24 h and analyzed by Western blot. **d, e** Assessment of USP7 degradation efficiency. Panc89 (**d**) and Ma-Mel-47 (**e**) cells were treated with increasing concentrations of PROTACs. **f, g** Quantification of USP7 degradation efficiency. USP7 levels in panels d + e were quantified by densitometry and half-maximal degradation concentrations (DC$_{50}$) of NK250 in Panc89 (**f**) and NK266 in

Ma-Mel-47 (**g**) cells were calculated from three biological replicates. Data are shown as mean ± s.d. ($n = 3$ independent experiments). **h, i** Determination of degradation kinetics. Panc89 (**h**) and Ma-Mel-47 (**i**) cells were treated with 1 μM of PROTACs for indicated times. **j, k** Degradation rescue experiments. Panc89 (**j**) and Ma-Mel-47 (**k**) cells were pretreated for 2 h with either DMSO, NEDDylation inhibitor MLN4924 (500 nM), proteasome inhibitor Carfilzomib (CFZ, 250 nM) or VHL ligand NK249 (10 μM) followed by 1 μM of NK250 (**j**) or NK266 (**k**) for 20 h. See Supplementary Fig. 2i for the chemical structure of NK249. **l** Orthogonal confirmation of USP7 depletion by immune fluorescence. Panc89 cells were treated with 1 μM NK250 for 24 h before staining for USP7 (green), actin (red), and with DAPI (blue). Scale bar: 10 μm. Uncropped blots are provided as Source Data.

%, while the D$_{max}$ of NK266 of 74 % indicated a less efficient target degradation. Treating cells for 24 h resulted in the same near complete degradation (D$_{max}$ = 90−92 %). This was accompanied by low DC$_{50}$ values of 4 nM and 24 nM for NK250 and NK266, respectively, demonstrating excellent degradation efficiency.

Notably, we observed a partial reduction of USP7 levels with both the negative control PROTAC NK245 and the inhibitor NK192 (Supplementary Figs. 3a, b). This phenomenon is consistent with the previously observed auto-regulation of USP7 ubiquitination[28,47], by which the DUB can self-regulate its abundance through auto-

deubiquitination. Importantly, PROTAC-mediated degradation depleted USP7 levels more potently (DC$_{50}$ and D$_{max}$) in all cell lines tested (Fig. 2c, d, Fig. 3b, c, Supplementary Figs. 3a, b). Equivalent assays with second-generation compounds showed NK225 and NK233 to mimic NK250 and NK266 in degradation efficiency and kinetics with lower potency (Supplementary Figs. 3a, b).

Next, we aimed to understand the molecular basis for the different degradation kinetics of cyclohexyl- (NK225, NK250) and cubane-containing (NK233, NK266) compounds. The formation of a stable ternary substrate-PROTAC-E3 ligase complex in cells is increasingly

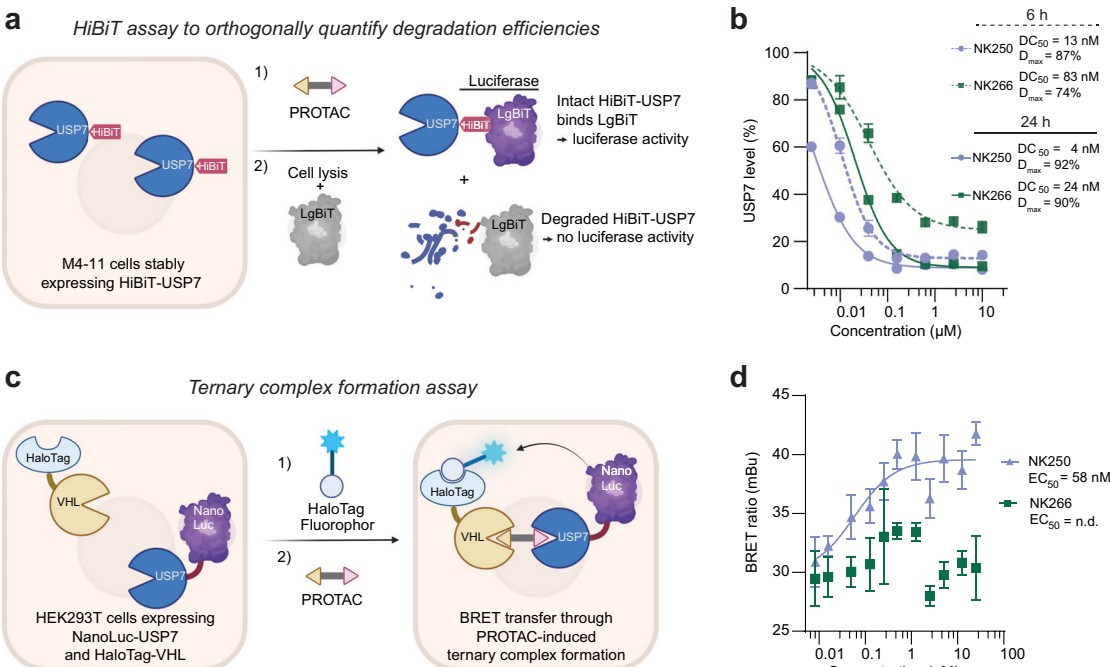

**Fig. 4 | Quantitative assessments of USP7 degradation efficiency and ternary complex formation. a** Schematic representation of HiBiT endpoint degradation assay. Panels a and c were partly created in BioRender. Klink, N. (2026) BioRender.com/f26vb8v. **b** HiBiT-based quantification of USP7 levels. MV4-11 cells stably expressing HiBiT-USP7 generated through lentiviral transduction were treated with PROTACs for either 6 or 24 h. Remaining HiBiT-USP7 could be detected through the luciferase signal using HiBiT Lytic Detection System. Data are shown as mean ± s.d. (n = 3 technical replicates), normalized to DMSO treatment. Half-maximal degradation concentrations ($DC_{50}$) derived from these data are given. **c** Schematic representation of a cellular ternary complex formation assay. Halotag-VHL- and NanoLuc-USP7-expressing cells are sequentially treated with Halotag-Fluorophor and PROTAC. Compound-induced proximity is read out by a bioluminescence resonance energy transfer (BRET) signal as shown, demonstrating ternary complex formation. **d** Cellular ternary complex formation assay. HEK293T cells over-expressing Halotag-VHL and NanoLuc-USP7 were treated with HaloTag NanoBRET 618 Ligand for 20 h, followed by NK250 or NK266 treatment at indicated concentrations for 2 h prior BRET measurement. The determined half-maximal ternary complex formation concentration ($EC_{50}$) for NK250 is given. Data are shown as mean ± s.d. (n = 4 technical replicates). mBu, milli BRET units.

recognized as a central paradigm for rapid targeted protein degradation[48]. Therefore, we employed a cellular ternary complex formation assay to determine the ability of PROTACs to induce proximity between USP7 and the VHL E3 ligase (Fig. 4c). We over-expressed both NanoLuc-USP7 and HaloTag-VHL in HEK293T cells and treated them with both a HaloTag-fluorophore and our PROTACs. Upon induction of a ternary complex, proximity between both pro-teins was recorded by measuring the resulting bioluminescence reso-nance energy transfer (BRET) from the NanoLuc luciferase to the dye associated with HaloTag-VHL[49]. In this assay, only NK250 and NK225 were able to potently induce a stable ternary complex between the PROTAC, USP7 and VHL with half maximal effective concentrations ($EC_{50}$) of 58 nM and 144 nM, respectively, while treatment with NK266 or NK233 did not increase the BRET signal to the same level (Fig. 4d, Supplementary Fig. 3c). Further, we verified that neither the negative control NK245 nor the VHL ligand VH298 induced a ternary complex. This potent induction of a stable ternary complex by NK250 provided a rational for the more rapid USP7 degradation facilitated by NK250 within 6 h versus 16–24 h by NK266 as shown both in Panc89 and MV4-11 cells. Collectively, these assays validated NK250 and NK266 as potent degraders of USP7, which enable the formation of validated and cell line-customized USP7 inhibitor and degrader pairs to interrogate cellular roles of USP7 in PDAC and melanoma.

### Proteome-wide analysis of USP7 degradation

Having established a chemical toolbox consisting of USP7-specific inhibitor and PROTAC pairs with chemical complementarity, we set out to examine consequences of USP7 modulation by directly com-paring USP7 inhibition and degradation. For this we treated both

Panc89 and Ma-Mel-47 cells with NK192 (5 μM) and PROTACs NK250/NK266 (1 μM) for 6, 24 or 72 h and recorded changes in the cellular proteome by mass spectrometry. Analysis of the Panc89 samples through data-dependent acquisition (DDA) on an Orbitrap Fusion Lumos mass spectrometer resulted in quantitation of 4214 unique protein groups (Supplementary Figs. 4a–f, Supplementary Table 1, Supplementary Data 4). The experiment demonstrated the exquisite specificity of NK250 for USP7 degradation as USP7 was the only protein significantly decreased after 6 h (Supplementary Fig. 4a). Moreover, USP7 protein levels decreased further in a time-dependent manner, leading to a more than 100-fold reduction after three days (Supple-mentary Figs. 4b, c). In contrast, USP7 inhibitor NK192 did not lead to USP7 protein level changes beyond the significance threshold (Sup-plementary Figs. 4d–g). However, we were surprised to observe only very few other proteins changing in their abundance within 24 h, including the previously validated USP7 substrate TRIP12[15].

To enable greater coverage, we thus adopted a data-independent acquisition (DIA) proteomics strategy on the same spectrometer[50]. Re-analysis of the same Panc89 samples with a DIA protocol allowed the identification and quantitation of 8828 unique protein groups (Fig. 5a–f, Supplementary Table 2, Supplementary Data 2), thereby more than doubling the return of the DDA analysis. Analysis of the Ma-Mel-47 cell samples using DIA yielded quantitative information on 9344 proteins (Fig. 5g–l, Supplementary Table 2, Supplementary Data 3), demonstrating a comprehensive coverage of the cellular proteome in both cell lines. These DIA data were then analyzed further (Fig. 5a–l).

As observed in the Panc89 DDA dataset, the DIA datasets revealed significant and consistent USP7 depletion upon PROTAC treatment in

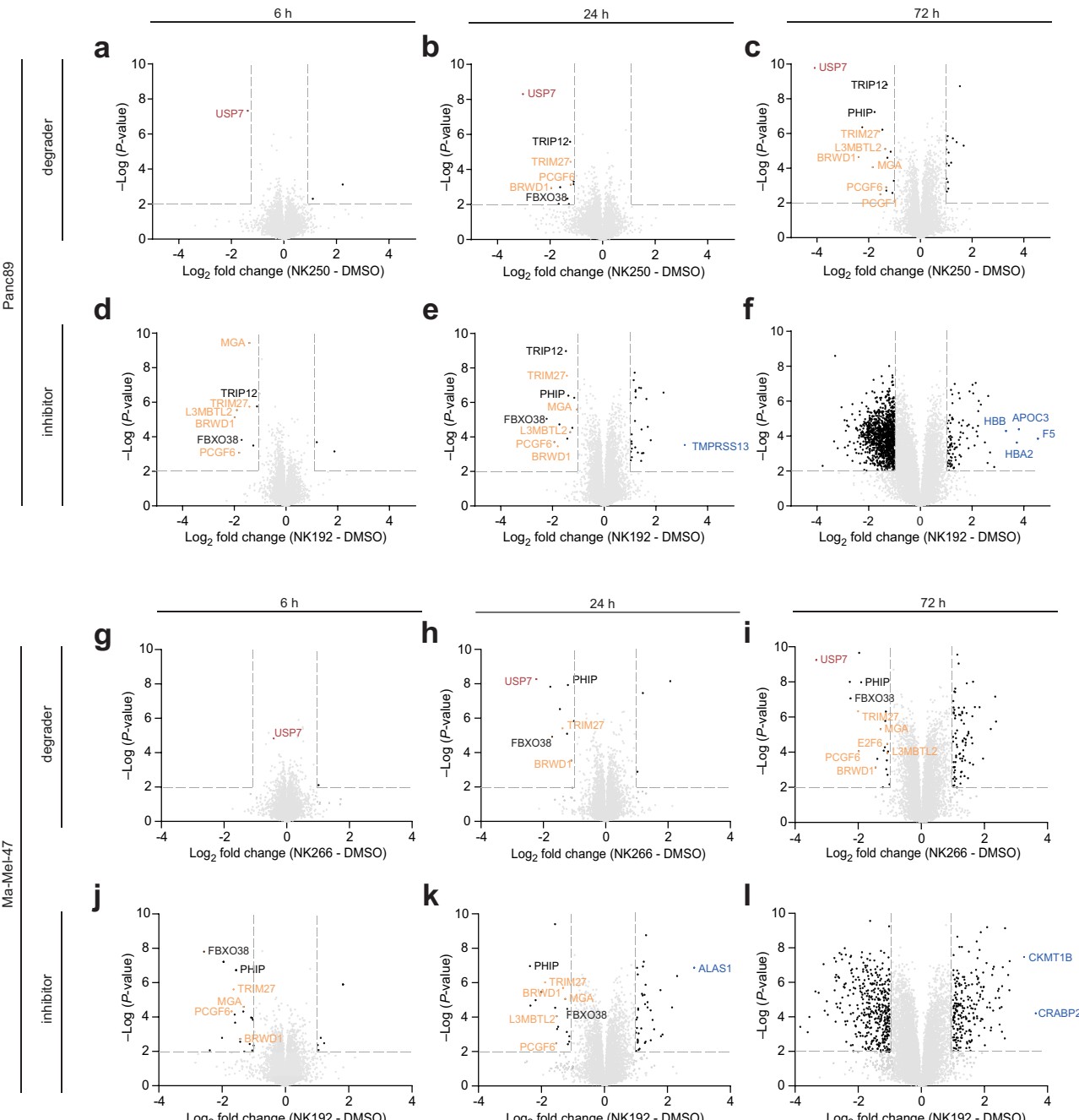

**Fig. 5 | Proteome-wide analysis reveals strikingly distinct effects of USP7 modulation through inhibitor versus degrader. a–f** Proteomic analysis of Panc89 cells treated with PROTAC NK250 (**a–c**) or inhibitor NK192 (**d–f**) for indicated time points. Volcano blots report proteins identified by data-independent acquisition mass spectrometry in Panc89 cells treated with NK192 (5 μM) or NK250 (1 μM) for 6, 24 or 72 h. Proteins annotated as members of non-canonical Polycomb repressive complexes are highlighted in orange, chromatin bound E3 ligases are highlighted in black, the most upregulated proteins by USP7 inhibition in blue. *P*-values were determined from two-sided *t*-tests as implemented in Perseus. **g–l** Proteomic analysis of Ma-Mel-47 cells treated with PROTAC NK266 (**g–l**, 1 μM) or inhibitor NK192 (**j–l**, 5 μM) for indicated time points.

both Panc89 and Ma-Mel-47 cells (Supplementary Fig. 4g). USP7 was the only protein that was significantly decreased in Panc89 cells after 6 h of NK250 treatment, further highlighting the specificity of our degrader (Fig. 5a). In accordance with the observations made through Western blot analysis, USP7 exhibited only minimal reduction in Ma-Mel-47 cells following 6 h of NK266 treatment (Fig. 5g). After 24 h, we identified USP7 to be the most decreased protein, thereby showing that also NK266 potently and specifically, yet more slowly, degrades USP7 in melanoma cells (Fig. 5h).

Moreover, we identified several established USP7 substrates to be decreased in their abundance following PROTAC treatment in both cell lines (Fig. 5a–c, g–i), including chromatin-bound E3 ligases (TRIP12, TRIM27)[15], members of the non-canonical Polycomb repressive complex 1.6 (ncPRC1.6) (MGA, L3MBLT2, BRWD1, PCGF6)[14,51] and diverse other proteins including a helicase (DHX40) and E3 ligase substrate adapters (FBXO38, PHIP)[27,51].

To understand the functional impact of these changes, we analyzed cells for phenotypes. We orthogonally validated our proteomic

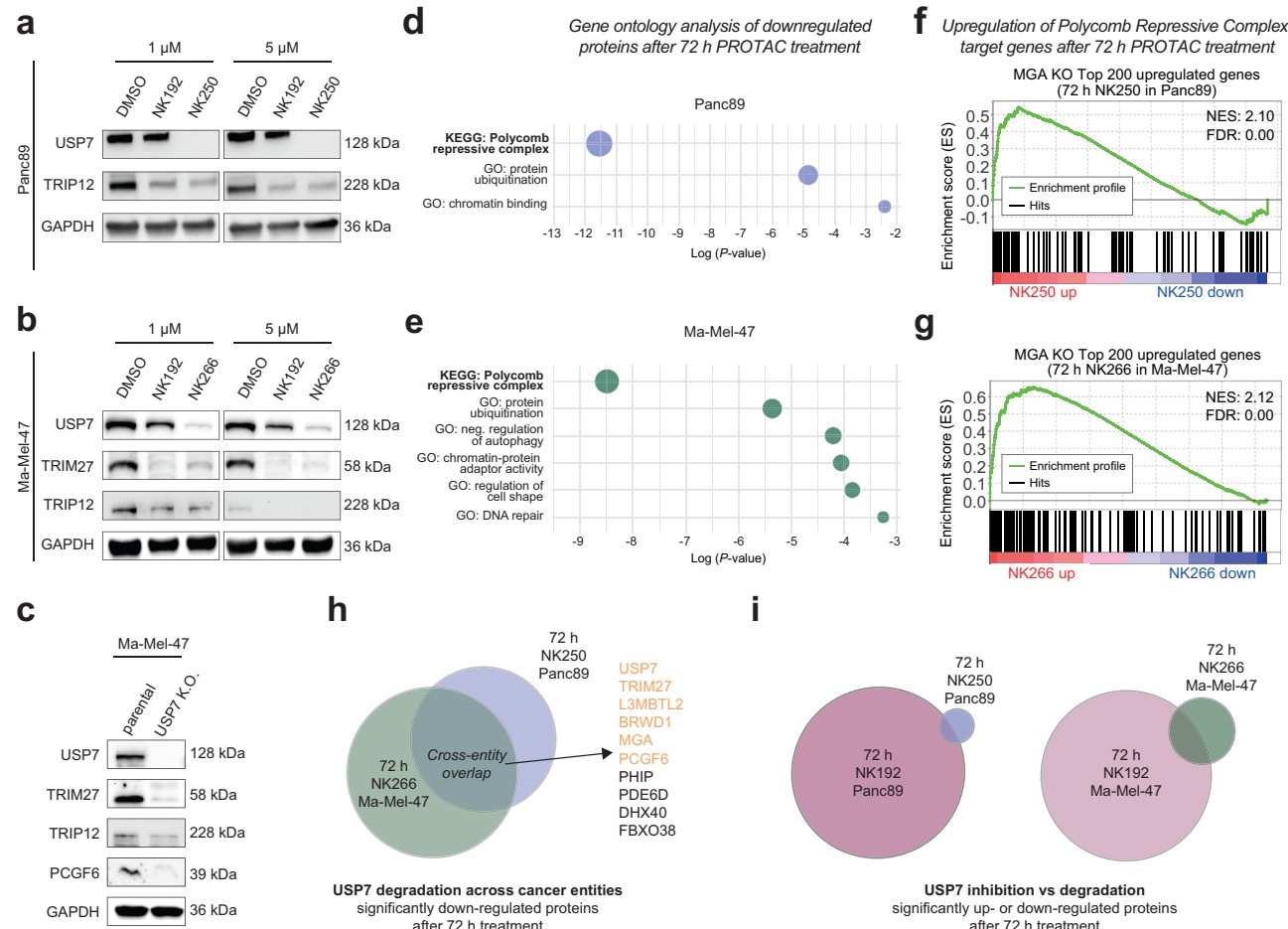

**Fig. 6 | Destabilization of non-canonical Polycomb repressive complex 1.6 subunits by selective USP7 degradation leads to de-repression of PRC target genes. a, b** Validation of proteomics results by Western blot analysis of USP7-regulated proteins identified by mass spectrometry. Panc89 (**a**) or Ma-Mel-47 (**b**) cells were treated with indicated compounds for 72 h. **c** Orthogonal on-target validation by genetic perturbation. Western blot analysis of USP7-regulated proteins in a Ma-Mel-47 knockout (K.O.) cell line. Blots were recorded from the same samples run on two separate gels. **d, e** Metascape pathway enrichment analysis of significantly downregulated proteins following 72 h of PROTAC treatment of Panc89 (**d**) or Ma-Mel-47 (**e**) cells (based on data shown in Fig. 5). **f, g** RNA-seq analysis as phenotypic readout after chemical USP7 depletion by PROTACs. Gene set enrichment analysis (GSEA) of RNA-seq data from Panc89 cells treated with

NK250 (**f**) or Ma-Mel-47 cells treated with NK266 (**g**), respectively, at 1 μM for 72 h. The enrichment of a PRC1.6 repressed gene set comprising the top 200 genes upregulated upon knock-out of the PRC1.6 component MGA from public expression data[53] was tested. **h** Venn diagram of significantly downregulated proteins after treatment of Panc89 cells with NK250 or Ma-Mel-47 cells with NK266 after 72 h (data in Fig. 5). ncPRC1.6 complex components are highlighted in orange. **i** Venn diagrams of significantly up- or downregulated proteins after 72 h of NK250 treatment in Panc89 (left) and NK266 treatment in Ma-Mel-47 (right) cells compared to 72 h of NK192 treatment. Cut-offs for significance (−Log (P-value) > 2) and fold change (log₂ (condition vs control) > (-)1) were used as shown in corresponding volcano plots in Fig. 5. Uncropped blots are provided as Source Data.

data by Western blot analysis and confirmed regulation of some of the most significantly downregulated proteins in Panc89 and Ma-Mel-47 cells (Fig. 6a, b). Next, we generated a Ma-Mel-47 USP7-knockout (USP7-KO) cell line and genetically confirmed that these substrates are indeed stabilized by USP7 (Fig. 6c). Pathway enrichment analysis of proteins reduced upon USP7 degradation identified the Polycomb repressive complex as most prominently downregulated across both cancer entities (Fig. 6d, e). Notably, two additional pathways, chromatin binding in Panc89 cells and chromatin-protein adapter activity in Ma-Mel-47 cells, were also downregulated, in line with the recently emphasized role of USP7 on chromatin regulation and gene expression[52].

To investigate cellular consequences of USP7-mediated destabilization of ncPRC1.6 subunits, we examined whether PROTAC-induced USP7 depletion leads to de-repression of genes normally silenced by ncPRC1.6 under basal conditions. Therefore, we performed RNA-seq analysis of Ma-Mel-47 and Panc89 cells after 72 h of PROTAC treatment. Genes repressed by the ncPRC1.6 complex were defined from a

public dataset generated after knockout of the ncPRC1.6 subunit MGA[53]. Using these data for gene set enrichment analysis (GSEA), we observed strong enrichment of ncPRC1.6-repressed genes among the most upregulated genes following USP7 degradation (Fig. 6f, g).

The limited number of regulated proteins shared by both cell lines (Fig. 6h), the majority of which are previously reported substrates of USP7, was unexpected given the near-complete depletion of this DUB, its multiple cellular roles and its ability to broadly regulate protein ubiquitination and protein stability[14,19,20,27,28,47]. Importantly, the lack of regulation in these cell lines of several previously reported substrates (including MDM2 and FOXO4, which gave rise to USP7 modulation to be pursued as a broadly applicable anti-cancer strategy[13,19,20,22,29,40]), as well as the large number of proteins changed only in one of the two investigated cell lines strongly hint at pronounced cell-line-specific differences in USP7 functions. Our findings thereby underscore the value of selective depletion of USP7 to investigate substrates and downstream effects in the cellular system of interest.

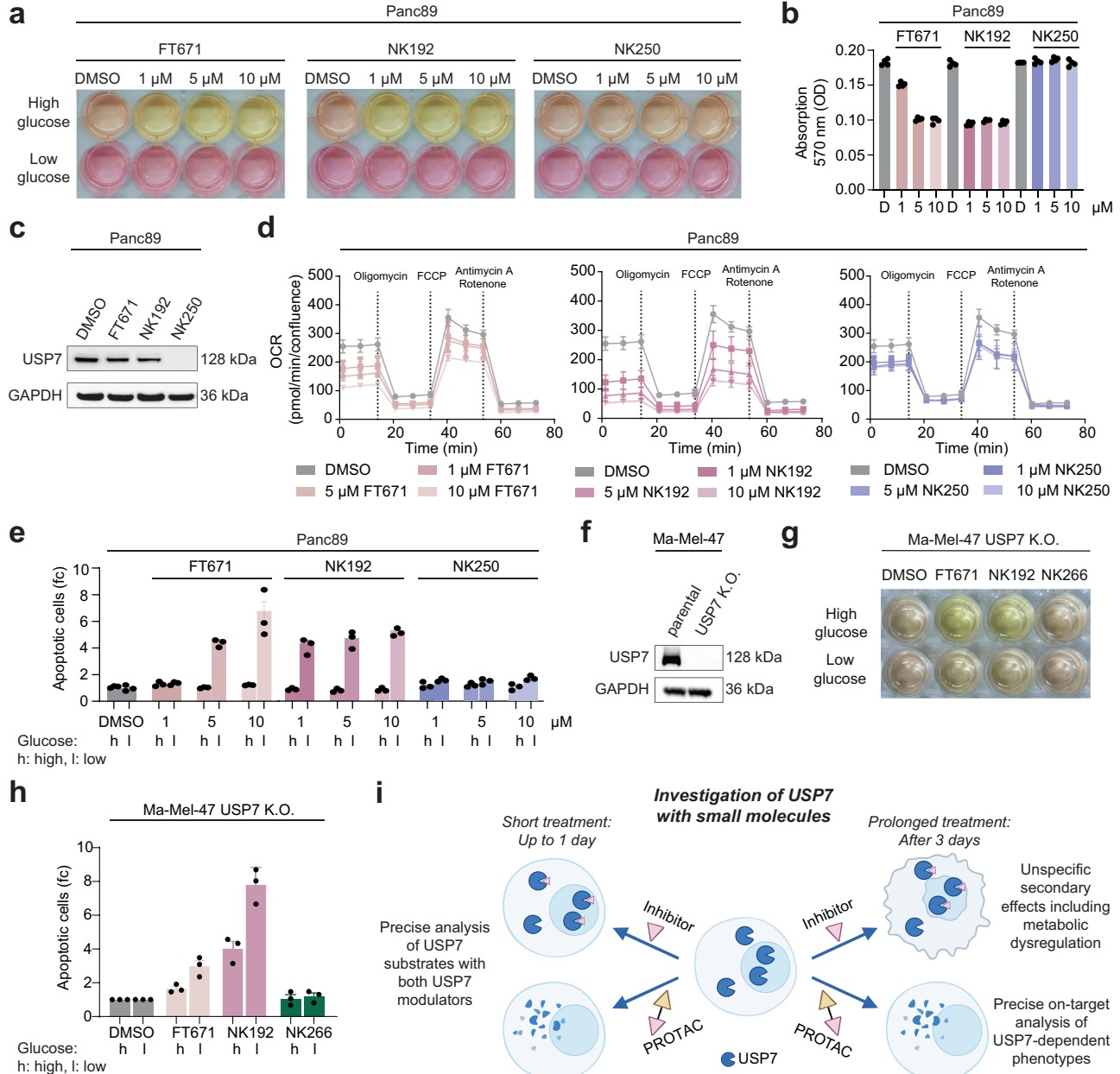

**Fig. 7 | Disparate metabolic effects of USP7 inhibitor vs. PROTAC treatment.**
**a** Images of Panc89 cells in culture media containing 5 mM (low) or 25 mM (high) glucose, treated with indicated concentrations of FT671, NK192 or NK250 for 72 h. **b** Photometric analysis of cell culture media of Panc89 cells treated as shown in a. Data are shown as mean (*n* = 3 technical replicates) from one of three biological replicates. D, DMSO. **c** Western blot analysis of Panc89 cells treated with 1 μM of indicated compounds for 72 h. **d** Seahorse mito stress test. Panc89 cells treated with indicated concentrations of FT671, NK192 or NK250 for 24 h, followed by measurements of their oxygen consumption rate (OCR). Three independent experiments were performed, in which at least triplicates were used for data analysis; one representative is shown as mean ± s.e.m. (*n* = 4-5 technical replicates). **e** Apoptosis assay. Panc89 cells treated with compounds for 72 h in culture medium containing 5 mM (low) or 25 mM (high) glucose were analyzed by FACS for Annexin-

V staining. Data were normalized to untreated controls; one representative experiment of at least three independent experiments, each with technical triplicates, is shown. Data are shown as mean ± s.e.m. (*n* = 3 technical replicates). fc, fold change. **f** Western blot analysis of genetic USP7 knockout of Ma-Mel-47 cells. **g** Images of Ma-Mel-47 USP7 K.O. cells in culture medium containing 5 mM (low) or 11 mM (high) glucose, treated with 5 μM NK192, 10 μM FT671 and 1 μM NK266 after 72 h treatment. **h** Apoptosis assay. Ma-Mel-47 USP7 K.O. cells treated with 5 μM NK192, 10 μM FT671 and 1 μM NK266 for 72 h in culture medium containing 5 mM (low) or 11 mM (high) glucose were analyzed as described in (**e**). **i** Schematic of proposed use of USP7 modulators for the investigation of USP7 with small molecules. Partly created in BioRender. Klink, N. (2026) BioRender.com/f26vb8v. See Supplementary Fig. 7g for the FACS gating strategy. Uncropped blots are provided as Source Data.

## Proteome-wide analysis reveals strikingly distinct effects of USP7 modulation through inhibition versus degradation

Treatment of Panc89 or Ma-Mel-47 cells with USP7 inhibitor NK192 for 6 h or 24 h resulted in similar sets of proteins being regulated as observed with USP7-degrader-treated cells (Fig. 5d, e, j, k). These findings robustly highlight the role of USP7 in regulating the

abundance of several E3 ligase proteins as well as of PRC1.6 complex members as a relatively small set of core substrates.

Strikingly, inhibitor treatment for 72 h induced drastic differences in the proteomes of Panc89 and Ma-Mel-47 cells. We found hundreds of proteins being either up- or downregulated (Fig. 5f, l), including an aberrant induction of several proteins not normally expressed in PDAC

or melanoma cells (e.g., the hemoglobin subunits HBB and HBA2). This finding contrasts with the comparably low number of proteins affected by PROTAC treatment across all time points. The large-scale, significant proteomic rewiring consistently observed in both cancer types after 72 h of inhibitor treatment, but not with PROTAC treatment, highlights the value of targeted protein degradation.

To confirm that these observed differences do not occur due to varying concentrations between inhibitor and degrader, we performed proteomic experiments using matched inhibitor and degrader concentrations (Supplementary Fig. 5a–j) for 72 h via the DIA proteomics strategy (Supplementary Table 3, Supplementary Data 5-6). Indeed, increasing PROTAC concentrations to 5 µM or reducing the inhibitor concentration to 1 µM showed proteomic responses in both Panc89 and Ma-Mel-47 highly similar to the data recorded at 1 µM PROTAC or 5 µM inhibitor, respectively. These data confirmed selective depletion of USP7 using PROTACs, contrasted with proteomic dysregulation by the inhibitor after prolonged treatment. Additionally, we investigated the cellular responses of our negative control PROTAC NK245 in both cell lines, which only showed minimal changes to the proteome after 72 h of treatment (Supplementary Fig. 5e, j), confirming the robustness of our chemical toolbox.

Pathway enrichment analysis of proteins significantly downregulated after 72 h (Fig. 5f, l) demonstrated that inhibitor treatment affected markedly more pathways than PROTAC treatment in both cell lines (Supplementary Fig. 6a–d, Fig. 6d, e). Notably, metabolic pathway regulation was observed in Panc89 cells, whereas cell cycle programs were affected in Ma-Mel-47 cells. In contrast, among proteins downregulated after 24 h of inhibitor treatment, only a limited number of pathways were affected, underscoring the suitability of inhibitors for investigating USP7 at short treatment durations (Supplementary Figs. 6c, d). Here, transcriptional regulation by E2F6 (a transcriptional repressor as part of the ncPRC1.6 complex[53]) emerged as the top enriched pathway in both entities, further supporting a central role for USP7 in ncPRC1.6-mediated transcriptional repression. The magnitude of proteomic changes following 72 h of inhibitor treatment indicates dysregulation arising from USP7-independent pleiotropic effects at extended treatment durations. In contrast, PROTAC treatment for the same duration only showed small and selective proteomic changes of mostly known USP7 substrates in both cancer entities (Fig. 6i).

## PROTAC-induced degradation of USP7 uncovered USP7-independent glucose dependency induced by USP7 inhibitors

Our proteomic data demonstrated that USP7 degraders caused relatively small proteomic changes, whereas broad proteomic remodeling was induced by the inhibitor after 72 h (Fig. 6i). This suggested that cells treated with USP7 inhibitor NK192 for prolonged times should show phenotypes distinct from USP7 depletion by PROTACs. Indeed, when various PDAC and melanoma cell lines were treated with USP7 modulators, we observed a strong switch in the cell culture media color for the inhibitor-treated, but not for the degrader-treated samples. To investigate this disparity systematically, we treated Panc89 cells with 1-10 µM of USP7 inhibitors FT671, NK192 and the PROTAC NK250 in standard complete DMEM medium containing either 25 mM ("high glucose") or 5 mM ("low glucose") D-glucose (Fig. 7a). In addition, cell growth was recorded after 72 h which was not affected in all conditions (Supplementary Fig. 7a). Interestingly, the color change of the culture medium was observed only under high-glucose conditions, but not in low-glucose media. This suggests acidification resulting from increased nutrient consumption in inhibitor-treated cells, leading to a pH decrease that alters the absorption spectrum of the pH indicator. The same effect was observed in Ma-Mel-47 cells growing in more lightly colored RPMI 1640, when the standard media conditions with 11 mM D-glucose were compared to those with 5 mM D-glucose (Supplementary Fig. 7b). Therefore, we quantified changes in absorption at 570 nm (a pH-sensitive absorption maximum of Phenol red) and

observed a lowered absorption of the standard cell media after inhibitor-, but not PROTAC treatment (Fig. 7b, Supplementary Fig. 7c). Complete depletion of USP7 in PROTAC-treated cells within 72 h for both Panc89 (Fig. 7c) and Ma-Mel-47 (Supplementary Fig. 7d) confirmed that the observed phenotypic disparity was not due to insufficient USP7 degradation.

To dissect this inhibitor-induced metabolic effect, we quantified cellular oxygen consumption rates (OCR) by seahorse assays. Whereas a degrader-mediated effect on OCR was much smaller and plateaued already at low doses where USP7 was fully degraded (Fig. 7d, Supplementary Fig. 7e), USP7 inhibitors facilitated a persistent reduction of mitochondrial OCR in a dose-dependent manner. This reduced capacity of ATP production through oxidative phosphorylation is consistent with a more anaerobic glucose metabolism in inhibitor-treated cells. Next, we tested if this change of metabolism would impact cell viability in a glucose-deprived environment. Inhibitor treatment, but not PROTAC treatment, in low glucose media strongly induced apoptosis in Panc89 (Fig. 7e) and Ma-Mel-47 cells (Supplementary Fig. 7f). These data indicated a higher glucose dependency or glucose consumption only due to inhibitor treatment, likely independent of USP7. This metabolic switch in turn explains the glucose-dependent acidification seen in culture medium (Fig. 7a, b, Supplementary Fig. 7b, c), whereas specific USP7 degradation upon PROTAC treatment in both PDAC and melanoma cells, through retained respiration capacity, did not induce glucose dependency (Fig. 7e, Supplementary Fig. 7f).

To confirm that this dependency was independent of USP7, we used our generated Ma-Mel-47 USP7-KO cell line (Fig. 7f) and investigated the effect of USP7 modulators. We detected the same change of media color with USP7 inhibitors FT671 and NK192, but not degrader NK266, as observed for the parental cells (Fig. 7g). Consequently, USP7-KO cells showed a drastic increase of apoptotic cells after inhibitor, but not degrader treatment. We thus conclude that inhibitor-induced metabolic and apoptotic changes are indeed USP7-independent effect of USP7 inhibitors NK192 and FT671 (Fig. 7h).

Taken together, these data highlight the value of highly specific USP7 degraders for distinguishing genuinely on-target USP7-mediated effects from non-specific, metabolic phenotypes induced even by well-characterized inhibitors (Fig. 7i). These findings thus have important implications for the cellular use of USP7 inhibitors FT671 and NK192, particularly with prolonged treatment times. More broadly, our data showcases the advantages of the PROTAC approach for the dissection of cellular roles of a key deubiquitinase through providing highly specific degraders.

## Discussion

The development of matched inhibitor-degrader pairs marks a critical step toward dissecting the multifaceted cellular roles of deubiquitinating enzymes (DUBs). Here, we present a comprehensive chemical biology framework for the interrogation of USP7 through PROTAC degraders and inhibitors. This dual-modality toolkit enables temporal and mechanistic analysis of USP7 functions across cancer types, while mitigating some of the limitations of classical inhibitor-based approaches.

VHL-based PROTACs NK250 and NK266 demonstrated potent and near-complete USP7 degradation in PDAC and melanoma cells, respectively. Degradation kinetics correlated with ternary complex formation, and the choice of VHL as E3 ligase proved strategic: unlike ligands for the Cereblon ligase[54], ligands for VHL do not induce activation of the p53-p21 axis. Given that USP7 activity regulates p21 in some cell lines[19,29], CRBN-induced neosubstrate degradation could confound mechanistic interpretation. Such possibly confounding effects are thus excluded through the design of VHL-based degraders[23,39].

A noteworthy observation was the modest reduction of USP7 protein levels upon prolonged inhibitor treatment, highlighting an

inherent autoregulatory mechanism. This complicates attempts to ascribe phenotypes solely to catalytic inhibition, particularly beyond 24 h of compound treatment where enzymatic blockade and protein loss co-occur. Such effects will need to be taken into account when distinguishing catalytic from scaffolding functions, for which our rapid degraders can be employed in future studies.

The proteomics data further illuminated the consequences of USP7 perturbation by inhibitor and degrader compounds. While early timepoints revealed selective depletion of known USP7 substrate proteins, prolonged inhibitor treatment triggered expansive proteomic remodeling, concomitant with USP7-independent metabolic rewiring. In contrast, independent of used concentrations, degraders produced narrower, on-target responses with limited changes to the proteome, supporting their specificity and utility in defining bona fide USP7-dependent pathways. This distinction is particularly relevant when revisiting phenotypes attributed to legacy inhibitors, such as P5091 or P22077, now known to potently engage other DUBs in cells like USP10 and USP47[11,38].

Notably, comparative proteomics across PDAC and melanoma revealed substantial cell-type-specific differences in USP7 substrate regulation. These findings reinforce emerging work suggesting that DUB functions are shaped by cell context and cell state[55]. The exploration of USP7 as a drug target must therefore consider these contextual nuances when evaluating compound efficacy or potential resistance mechanisms.

In phenotypic assays, both NK250 and NK266 slightly reduced oxygen consumption rates, supporting a role for USP7 in bioenergetics. However, hydroxypiperidine-based inhibitors, such as NK192 and FT671 at prolonged treatment times induced metabolic collapse, glucose dependency, and widespread proteomic disturbances, even in the genetic absence of USP7. Thereby, we could attribute the extent and intensity of these phenotypes to off-target engagement of the inhibitors. Additionally, PRC complex perturbation in a USP7-inhibited state may contribute[14,56]. These findings highlight the importance of systematic chemical compound validation and the superior fidelity of degrader-based approaches, particularly for mechanistic studies with longer compound treatment times.

Taken together, our results support a strategy for comprehensively interrogating the cellular roles of USP7 using small molecules (Fig. 7i). Targeted USP7 degradation using our PROTAC toolbox achieves highly selective modulation of downstream cellular processes across both cancer entities (Fig. 6h) and all analyzed time points. Selective inhibitors remain valuable for rapid interrogation of catalytic function and substrate identification. However, our data highlight an important caveat when using hydroxypiperidine-based USP7 inhibitors, such as NK192 or FT671 for prolonged exposure, as these show cellular phenotypes independent of USP7. In these extended treatment times, high-fidelity degraders provide superior specificity and mechanistic resolution including functional dissection beyond catalysis. The complementary application of these tools, as demonstrated here, will enable nuanced investigation of USP7's catalytic and non-enzymatic roles while minimizing confounding artifacts.

In summary, we introduce a rigorously characterized inhibitor-degrader pair for USP7 that supports mechanistically resolved and time-adapted experimental design. These tools will not only refine our understanding of USP7 in cancer biology but also exemplify the broader potential of degrader-inhibitor complementarity in unraveling the complexity of the ubiquitin system.

## Methods
### Chemicals and chemical synthesis
Synthetic procedures and analytic data for compounds prepared for this work are shown in the Supplementary Information (Supplementary Figs. 8–18). Chemicals, which were not synthesized for this work, were purchased from MedChemExpress (FT671: #HY-107985; XL177A:

#HY-138793; USP7-In-3: #HY-112128; P5091: #HY-15667; GNE-6640: #HY-112937; Carfilzomib (CFZ): #HY-10455 and Pevonedistat/MLN4924: #HY-70062). All compounds (inhibitors, PROTACs and controls) were dissolved in DMSO, which was used as vehicle control.

### Cell culture
The melanoma cell line Ma-Mel-47 (established previously[57] from metastatic lesions from patient tumor tissues after written informed consent) and the PDAC cell line Panc89 (RRID: CVCL_4056, also referenced as T3M-4[58]) were cultured in a humidified incubator with 5% $CO_2$ at 37 °C in RPMI1640 (Ma-Mel-47) / DMEM (Panc89) medium containing 10% (v/v) fetal calf serum (FCS) and penicillin/streptomycin (all Gibco, Thermo Fisher Scientific). Studies on human material were approved by the institutional review board. Cells were confirmed to be negative for mycoplasma contamination. Testing for mycoplasma was performed by PCR monthly. To confirm cell identity, STR analysis was performed by the Microsynth Seqlab GmbH.

### Creating stable USP7 knockout cells by CRISPR/Cas9 editing
Ma-Mel-47 cells were co-transfected with HAUSP CRISPR KO and HAUSP HDR plasmids obtained from Santa Cruz Biotechnology (sc-402013-KO-2; sc-402013-HDR-2). To this end, 200,000 cells were seeded in a 6-well-plate and transfected 24 h later with 2 μg DNA (1 μg KO plasmid plus 1 μg HDR plasmid) using jetPRIME transfection reagent and buffer (Polyplus Sartorius) according to manufacturer's protocol (in 2 mL RPMI 1640 medium for 48 h). HAUSP HDR plasmids are a mixture of two to three plasmids, each encoding a target-specific template for homologous recombination at the cleavage sites generated by the HAUSP CRISPR/Cas9 KO plasmid. In addition, HDR plasmids carry a puromycin resistance cassette for the selection of stable knockout cells and an RFP marker to enable visual confirmation of transfection. Following transfection, cells were selected via puromycin treatment with 1.5 μg/mL puromycin for 72 h and until complete demise of control cells. Puromycin-resistant cells were seeded as single clones by limiting dilution into 96-well plates and clonally expanded. Cell pellets were harvested at three different time points and analyzed for USP7 protein levels by Western blotting.

### PROTAC rescue experiments
Cells were treated with proteasome inhibitor Carfilzomib (CFZ, 250 nM), NEDDylation Inhibitor MLN4924 (500 nM) or VHL inhibitor NK249 (10 μM). After 2 h, PROTAC (1 μM) was added. Cells were analyzed after 20 h.

### Preparation of whole cell extracts
To prepare samples for Western blot analysis, cells were seeded in 6-well dishes the day before treatment to adhere at 70−80% confluency. Compounds were added to the culture media at working concentrations of 1–10 μM and incubated between 2 and 72 h.

Cell pellets of all PDAC samples were lysed via adding a buffer containing 150 mM NaCl, 50 mM Tris/HCl (pH 7.4), 1% Triton X-100, 10 mM EDTA, 1 mM PMSF, 10 mM orthovanadate, and 1% aprotinin, incubating on ice for 1 h with vortexing every 15 minutes. Lysates were cleared of cell debris by centrifugation at 10,000 ×g at 4 °C for 10 min. Cell pellets of melanoma samples were lysed using Cell Lysis Buffer #9803 (Cell Signaling Technology) and incubating for 15 min on ice. Lysates were then clarified of cell debris by centrifugation at 13,000 ×g at 4 °C for 15 min. Thereafter, the total protein concentrations were either determined immediately or the protein extracts were stored at −20 or −80 °C. Pellets for samples shown in Fig. 6 and Supplementary Fig. 6 were lysed in 120 μL lysis buffer (50 mM Tris pH 8.0, 150 mM NaCl, 1% (v/v) IGEPAL, 0.5% (w/v) Na-deoxycholate, 0.1% (w/v) SDS) supplemented with 1x EDTA-free inhibitor protease cocktail (cOmplete, Roche) for 15 min at 4 °C. 0.2 μL Turbonuclease (Sigma Aldrich) and 5 mM MgCl2 were then added and lysates were incubated for

15 min at 4 °C. Cell debris was separated by centrifugation at 20,817 ×g for 15 min at 4 °C.

Protein concentrations of cell lysates were measured using the Pierce BCA Protein Assay (Thermo Fisher Scientific) or the Quick Start Bradford Protein Assay (Bio-Rad), following the manufacturers' protocols. Measurements were recorded on Tecan Spark or Tecan Infinite M200 plate readers. Lysates were kept on ice throughout the entire process.

## SDS-PAGE and western blotting

To separate proteins via SDS-PAGE, 10-20 μg of protein in lysis buffer was diluted with SDS-PAGE loading buffer (for most blots containing DTT or 2-mercaptoethanol) and denatured at 95 °C for 5 min. Protein separation was conducted at 95–110 V using the Mini-PROTEAN electrophoresis system (Bio-Rad). Proteins were then transferred to PVDF or nitrocellulose membranes using the Trans-Blot Turbo system (Bio-Rad) with the Trans-Blot Turbo RTA Kit for PVDF membranes. Prior to transfer, membranes were activated by soaking in 100% ethanol (for PVDF) or distilled water (for nitrocellulose) and proteins were transferred using the Mixed MW or the standard (1.0 A, 25 V, 30 min) protocol.

After transfer, membranes were blocked in 5% (w/v) milk/TBS-T for 1 h and then washed three times for 10 min with TBS-T. They were incubated overnight at 4 °C with the primary antibody in 5% (w/v) milk/TBS-T or 1% (w/v) BSA/TBS-T. Membranes were then washed three times for 10 min with TBS-T and incubated with horseradish peroxidase (HRP)-conjugated secondary antibodies in a solution of 5% (w/v) BSA/TBS-T or 1% (w/v) milk/TBS-T for 1 h at room temperature. PBS-T was used for blots shown in Fig. 6 and Supplementary Fig. 6.

Membranes were incubated with Clarity Western ECL substrate (Bio-Rad) for PDAC samples or Pierce ECL Western Blotting-Substrate (Thermo Fisher Scientific) for melanoma. Chemiluminescent signals were detected with a ChemiDoc Imaging System (Bio-Rad) for PDAC samples or with a ECL Chemostar Imager (Intas Science Imaging) for melanoma samples.

Antibodies were purchased from Cell Signaling Technology (USP7: HAUSP (D17C6) XP rabbit mAb #4833, 1:1000; GAPDH: GAPDH (14C10) rabbit mAb #2118, 1:10,000 or 1:5000; TRIM27: TRIM27 (D5S4O) rabbit mAb #15099, 1:1000; secondary: goat anti-rabbit IgG, HRP-linked #7074, 1:10,000), Santa Cruz Biotechnology (PCGF6: PCGF6 (A-6) mouse mAb #sc-518220, 1:100), Proteintech (TRIP12: rabbit polyclonal Ab #25303-1-AP, 1:500), Abcam (USP7: rabbit polyclonal Ab #ab190183, 1:2000), Thermo Fisher (GAPDH: GAPDH (6C5) mouse mAb #AM4300, 1:10000), Merck (secondary: sheep anti-mouse HRP-linked, NXA931, 1:10,000; secondary: donkey anti-rabbit HRP-linked, GENA934, 1:5000), and Jackson ImmunoResearch Labs (secondary: goat anti-mouse IgG HRP-linked #115-035-003, 1:5000).

Densitometric quantification of band intensities for Fig. 2c, d was performed using the ImageJ software. Densitometric quantification of band intensities for Fig. 3d, g and Supplementary Fig. 2b, e was performed using the Image Lab software. USP7 band intensities were normalized to GAPDH levels. DC50 values were calculated using the sigmoidal dose-response (three parameters) equation in GraphPad Prism and are given as the concentration where 50% of the total signal was remaining.

## Seahorse mito stress test and confluence measurements

The Seahorse XF cell mito stress test was used to measure oxygen consumption rates (OCR). Cells ($1.5×10^4$ per well) were seeded into cell culture microplates (Agilent) in culture medium containing either 1, 5 or 10 μM FT671, NK192, NK250 (Panc89), NK266 (Ma-Mel-47) or vehicle control 24 h prior to start of the assay. For normalization, cells were simultaneously seeded into 96-well plates ($1.2×10^4$ per well), treated with compounds, and cell confluence was measured on the day of the assay using the NYONE automated cell imager (Synentec). To reveal

key parameters of mitochondrial functionality, Oligomycin (1.5 μM), Carbonyl cyanide-4-trifluoromethoxyphenylhydrazone (FCCP; 1 μM), Rotenone (0.5 μM) and Antimycin A (0.5 μM) were loaded into ports of the XF sensor cartridge. After washing of the cells and calibration of the XF sensor cartridge plate, OCR values were determined by an XF96/XFe96 Seahorse Analyzer. Data analysis was conducted using the Wave software (Agilent).

## Annexin V staining and medium color switch

Annexin V staining was used to assess apoptosis of Panc89 and Ma-Mel-47 cells. Cells were seeded in a 12-well format and pre-treated with 1, 5 or 10 μM FT671, NK192, NK250 (Panc89), NK266 (Ma-Mel-47) or vehicle control for 24 h in standard culture medium conditions. Cells were then treated with the same compounds in high (DMEM: 25 mM; RPMI: 11 mM) or low (5 mM) glucose culture medium. After 72 h of compound treatment, cells were either harvested, stained with FITC Annexin V (1:30; BD Biosciences 556419) and DAPI (1:1000), and subsequently measured on a BD FACSCelesta Cell Analyzer (BD Biosciences) or a photo was taken to visualize the medium color switch. Data analysis of FACS data was performed with the FlowJo Software.

The Ma-Mel-47 USP7 knockout cell line apoptosis assay was performed similarly. Cells were treated with 10 μM FT671, 5 μM NK192, 1 μM NK266 or vehicle control in high (11 mM) or low (5 mM) glucose culture medium (RPMI-1640). After 72 h of compound treatment, cells were harvested, stained with Annexin V-APC (1:40, Thermo Fisher Scientific A35110) and propidium iodide (1:400 PI; BD Pharmingen Propidium Iodide Staining Solution 556463) and incubated at room temperature in the absence of light for 15 min. Afterwards, PBS was added, and cells were immediately measured on a Gallios flow cytometer from Beckman Colter. Analysis of FACS data was performed with the KaluzaAnalysis Software (Beckman Colter).

## Cellular ternary complex formation assay

The assay was performed as described step-by-step elsewhere[49]. USP7 (NanoLuc-USP7) and VHL (HaloTag-VHL) were cloned in frame with N-terminal NanoLuc or HaloTag, respectively. For transfections, HEK293T cells were diluted in Opti-MEM medium without phenol red (Life Technologies) to $4×10^5$ cells/mL and 9 μL were pipetted into each well of a white 384-well plate (Greiner 781207). 1 mL transfection mix was prepared containing 30 μL FuGENE HD (Promega, E2312) and 8 μL of each plasmid, incubated 20 min at RT and 1 μL were pipetted into each well leading to the total assay volume of 10 μL. After incubation at 37 °C and 5% $CO_2$ for 20 h, 40 nL HaloTag NanoBRET 618 Ligand (Promega) was added to the cells using an Echo acoustic dispenser (Labcyte) and the cells were incubated for an additional 20 h at 37 °C and 5% $CO_2$. 2 h prior to BRET measurements, the compounds were titrated to the cells using an Echo acoustic dispenser and the cells were incubated further at 37 °C and 5% $CO_2$ to allow complex formation. NanoBRET Nano-Glo substrate and Extracellular NanoLuc Inhibitor (Promega, N2540) were then added as per the manufacturer's protocol, and filtered luminescence was recorded on a PHERAstar FSX plate reader (BMG Labtech) equipped with a luminescence filter pair (donor: 450 nm BP filter; acceptor: 610 nm LP filter). Data were analyzed using GraphPad Prism 10 software.

## HiBiT USP7 degradation assay

**Cell culture.** Human MV4-11 and HEK293 cells were cultured at 37 °C in 5% $CO_2$ in RPMI-1640 and DMEM medium, respectively, each supplemented with 10% FBS and 1% penicillin/streptomycin.

**Cloning.** HiBiT-USP7 construct was cloned by extraction of USP7 fragment from plasmid pQHA-USP7 WT puroR (Addgene #46753) using appropriate restriction sites. The USP7 fragment was then ligated into the pRRL-PGK-HiBiT entry vector to generate the plasmid.

**Cell line generation.** Lentiviral infection was used to generate stable a MV4-11 HiBiT cell line. Lentivirus was produced using plasmids psPAX2, pMD2.G, and HiBiT-USP7 plasmid in HEK293 cells. MV4-11 cells were infected with filtered virus supernatant and selected after 48 h of infection for the generation of the stable MV4-11[HiBiT-USP7] cell line.

**HiBiT assay.** The assay was performed as described previously[6]. Briefly, MV4-11[HiBiT-USP7] cells were seeded and treated with compounds for the indicated time. Nano-Glo HiBiT Lytic Detection System (Promega) was used for the assay and luminescence was measured on a Tecan Spark microplate reader (Tecan). $DC_{50}$ values were calculated using the sigmoidal dose-response (four parameters) equation in GraphPad Prism and are given as the concentration where 50% of the total signal was remaining. $D_{max}$ values are given as the maximum degradation observed at any concentration.

## Immunofluorescence staining and microscopy

Panc89 cells were seeded on 96-well plates (89626, Ibidi) and treated with either DMSO or compound for the indicated time and concentration. Cells were then fixed with 4% PFA in PBS for 20 min, permeabilized with 0.1% Triton X-100 in PBS for 10 min and incubated in blocking buffer containing 2% BSA in PBS for 1 h. Three 10 min washes with PBS were performed between different steps. Cells were treated with primary antibody anti-USP7 (ab190183, Abcam, rabbit, 1:500) in blocking solution for 1 h at RT. After three washes with PBS, the fluorescently labeled secondary antibody (goat anti-rabbit IgG labeled with Alexa Fluor Plus 488, A32731, Invitrogen, 1:1000) in blocking buffer was incubated together with rhodamine-phalloidin (R415, Invitrogen, 1:1000) and HCS CellMask deep red (H32721, Invitrogen, 1:5000) for 1 h at room temperature. Finally, cells were stained with DAPI (Invitrogen, 1 μg/mL) in PBS for 10 min and mounted with aqueous mounting medium (50001, Ibidi) for microscopy. Plates were imaged on a fully automated inverted EVOS M7000 microscope system (Invitrogen) with X-Apo 20x air objective (0.8 NA, Olympus, AMEP4906) using DAPI (AMEP4950), GFP (AMEP4951), RFP (AMEP4952), and Cy5 (AMEP4956) LED light cubes. Images were captured using a high-sensitivity 3.2 MP monochrome CMOS camera and were processed with Omero (v. web 5.28.0).

## USP7 inhibition assay

USP7 inhibitors in DMSO were diluted in UbR buffer (40 mM Tris-HCl pH 8, 150 mM NaCl, 1 mM TCEP, 0.1 mg/mL BSA, 0.01% Tween20) to a concentration of 80 μM (4x). This stock was then further diluted in an 11-times serial dilution in UbR buffer containing 4% DMSO (4x). To this was added the same volume of human USP7 full length protein[59] (1.2 nM in UbR buffer, 4x). The resulting mixture was incubated for 15 min at rt after which 10 μL were added to a black 384 well low volume non-binding surface plate (Greiner 784900) in triplicates. The assay was initiated upon addition of 10 μL per well of Ubiquitin-Rhodamine110Gly (Ub-RhoG) (100 nM in UbR buffer, 2x). A TCEP-free UbR buffer was used for assaying P5091. Normalization was carried out using samples featuring either no inhibitor or no enzyme as respective controls. Fluorescence was measured on a Tecan Spark plate reader for 1 h in 2 min intervals at 25 °C (excitation = 492 nm, emission = 525 nm). Slopes of fluorescence over time curves were determined with Microsoft Excel through linear regression using the first 15 min of each measurement. Biochemical $IC_{50}$ values were calculated in GraphPad Prism 10.4.1 using non-linear regression (4 parameters dose response).

## 3' bulk mRNA sequencing

Ma-Mel-47 cells were exposed to 1 μM NK266 and Panc89 cells were exposed to 1 μM NK250 (or DMSO as a vehicle control) in each case for 72 h. Following treatment, cells were lysed and total RNA was isolated using the AllPrep DNA/RNA Kit (Qiagen) in accordance with the

manufacturer's instructions. RNA quality and integrity were evaluated using an Agilent Tape Station system. Library preparation was carried out following the QuantSeq 3' mRNA Library Prep Kit protocol (Lexogen) with 12-nt unique dual indices (UDI Set A1; Lexogen), including incorporation of unique molecular identifiers (UMIs) using the UMI Second Strand Synthesis Module for QuantSeq (Lexogen). For each sample, 500 ng of RNA was used as input. PCR amplification was carried out using the PCR Add-on and Reamplification Kit V2 (Lexogen) for 15–20 cycles, with the exact number of cycles determined individually by quantitative PCR. Library purification was performed using the Purification Module with Magnetic Beads (Lexogen). Library size distribution and concentration were evaluated using the Agilent TapeStation. Sequencing was performed by Lexogen GmbH (Austria) using their in-house sequencing service, which included library multiplexing, run-level quality control, and demultiplexing of reads. Demultiplexed FASTQ files were provided for downstream analysis.

## Data processing and analysis of 3' bulk mRNA sequencing

Sequencing quality was checked using FastQC v0.12.1. QuantSeq Adapters and the polyA tail were trimmed using cutadapt v1.18[60]. FASTQs were aligned to the GRCh38 reference genome using STAR v2.7.10b. Duplicate reads were removed based on unique molecular identifiers using UMI-tools v1.0.1[61]. BAM files were indexed using samtools v1.6. Gene-wise reads were counted using the GenomicAlignments package v1.38.2[62] in R v4.3.3 and differential gene expression analysis was performed using edgeR v4.0.16[63]. RNA-seq data of MGA knock-out in HEK293 cells was downloaded from the BioStudies entry E-MTAB-6005[53] and analyzed as described above, excluding UMI deduplication. The top 200 upregulated genes upon MGA knock-out were used to perform GSEAPreranked analysis with GSEA v4.3.3.

## Competitive pulldown assay

Panc89 cells were lysed in 5 mL of chemoproteomics lysis buffer (50 mM Tris pH 7.5, 150 mM NaCl, 1% NP-40 Alternative, 5% Glycerol, 2 mM TCEP) supplemented with 2 μL Benzonase on ice for 2 h. Lysate was cleared by centrifugation for 15 min at 17,000 ×g, supernatant was transferred into a fresh tube and kept on ice. Protein concentration was determined using a DC-Assay kit (Bio-Rad). In parallel, streptavidin magnetic beads (30 μL per sample, Pierce #88817) were saturated with 1.1 equiv. of NK264 for 1 h at 4 °C. Afterwards beads were washed three times using chemoproteomics washing buffer (10 mM Tris pH 7.5, 150 mM NaCl, 0.05% Tween20) and equally divided into tubes. Afterwards, Panc89 lysate (500 μL containing 700 μg of protein) containing either DMSO or NK192 (10 μM) were added onto the beads, and the mixture was inverted overnight at 4 °C. Next, the mixture was briefly centrifuged to gather the solution, beads were pelleted using a magnet and the supernatant was discarded. Beads were washed two times with chemoproteomics washing buffer and once with PBS. After each washing step, beads were pelleted using a magnet and the supernatant was discarded. Finally, bound proteins were eluted off the beads by addition of 50 μL of 2x SP3 buffer (2% SDS, 20 mM TCEP, 80 mM CAM, 100 mM Hepes pH 8.0) followed by heating to 90 °C for 5 min with mixing. Afterwards, beads were separated using a magnet and the supernatant was transferred into a fresh tube. This elution step was repeated by adding 50 μL of 1x SP3 buffer (1% SDS, 10 mM TCEP, 40 mM CAM, 50 mM Hepes pH 8.0) onto the beads and heating at 90 °C for 5 min. First and second eluates were united and frozen at −80 °C until further processing.

## Sample preparation for enriched proteome samples

3 μL per sample of a 50 μg/μL 1:1 mixture of hydrophilic (#45152105050250) and hydrophobic (#65152105050250) carboxylate modified Sera-Mag SpeedBeads (Cytiva), that were washed twice with MS-grade water, were added. The next steps were carried out at room temperature unless noted otherwise. Afterwards, the samples were

mixed shortly (1 min, 1000 rpm) and collected by short centrifugation (10 s, 200 ×g). Protein binding was induced by the addition of an equal volume of pure ethanol (10 min, 1000 rpm), the beads were collected by a brief centrifugation step (10 s, 200 ×g) and the plate was placed on a magnetic stand. Beads were allowed to bind for at least 5 min before the supernatant was removed. The beads were then taken up in 180 μL 80% ethanol and transferred to a fresh multiwell plate.

Subsequently the beads were washed four times with 180 μL 80% (v/v) ethanol prior to the addition of 100 μL digestion enzyme mix (0.6 μg of trypsin (V5111; Promega) and 0.6 μg LysC (125-05061; FUJI-FILM Wako Pure Chemical) in 25 mM ammonium bicarbonate). Samples were incubated at 37 °C for 19 h while shaking (1300 rpm). On the next day, the samples were briefly centrifuged (10 s, 200 ×g) and placed on a magnet for 5 min. The clear solution containing the tryptic peptides was transferred to a fresh multiwell plate. The beads were taken up in 47 μL 25 mM ammonium bicarbonate and incubated while shaking (10 min, 1000 rpm). The plate was then placed on a magnetic stand and after 5 min the cleared supernatant was collected and combined with the recovered first peptide mix, followed by the addition of formic acid (FA) to a final concentration of 2% (v/v) for trypsin inactivation.

## Sample preparation for total proteome analysis

Panc89 and Ma-Mel-47 cells were treated in quadruplicate with 1 μM NK192, 5 μM NK192, 1 μM NK245, 1 μM NK250, 5 μM NK250 (only PDAC), 1 μM NK266, 5 μM NK266 (only Melanoma), or DMSO for 6 h, 24 h or 72 h. Following treatment, cells were harvested, washed twice with PBS, and pellets were flash frozen in liquid nitrogen before being stored at −80 °C. Samples were prepared using the single-pot, solid-phase-enhanced sample-preparation (SP3) strategy[64]. All buffers and solutions were prepared with mass spectrometry (MS)-grade water (Honeywell). Cell pellets were taken up in 100 μL 1x sample buffer (50 mM Hepes pH 8.0, 1% (wt/v) SDS, 1% (v/v) NP-40, 10 mM TCEP, 40 mM chloroacetamide) and samples were heated at 90 °C for 5 min prior to sonication with a Bioruptor UCD-300 (Diagenode) for ten cycles of 1 min pulse and 30 s pause at high power. The samples were then supplemented with Benzonase (20 U/sample) and sonified for 5 min in a water bath (Bandelin) followed by 30 min incubation at 30 min.

The protein extracts were then centrifuged (20000 ×g, RT, 40 min) and the protein concentration of the cleared lysates was determined using the Pierce 660 nm Protein Assay Reagent (#22660; Thermo Scientific) with the Ionic Detergent Compatibility Reagent (#22663; Thermo Scientific) according to the manufacturers' instructions. Next, 15 μg of total protein per sample in a volume of 100 μL sample buffer was transferred to a 500 μL 96-well plate and treated with 7 U of benzonase (#71206; Merck Millipore) in dilution buffer (20 mM Hepes pH 8.0, 2 mM MgCl$_2$) at 37 °C for 30 min with agitation at 1500 rpm. After Benzonase treatment samples were heated to 90 °C for 5 min and after cooling down to room temperature, iodoacetamide to a final concentration of 10 mM was added for complete alkylation of cysteines (37 °C, 15 min, 1000 rpm, in the dark).

Then, 3 μL of a 50 μg/μL 1:1 mixture of hydrophilic (#45152105050250) and hydrophobic (#65152105050250) carboxylate modified Sera-Mag SpeedBeads (Cytiva), that were washed twice with MS-grade water, were added to the samples. The next steps were carried out at room temperature unless noted otherwise. Afterwards, the samples were mixed shortly (1 min, 1000 rpm) and collected by short centrifugation (10 s, 200 ×g). Protein binding was induced by the addition of an equal volume of pure ethanol (10 min, 1000 rpm), the beads were collected by a brief centrifugation step (10 s, 200 ×g) and the plate was placed on a magnetic stand. Beads were allowed to bind for at least 5 min before the supernatant was removed. The beads were then taken up in 180 μL 80% ethanol and transferred to a fresh multiwell plate.

Subsequently the beads were washed four times with 180 μL 80% (v/v) ethanol prior to the addition of 100 μL digestion enzyme mix (0.6 μg of trypsin (V5111; Promega) and 0.6 μg LysC (125-05061; FUJI-FILM Wako Pure Chemical) in 25 mM ammonium bicarbonate). Samples were incubated at 37 °C for 19 h while shaking (1300 rpm). On the next day, the samples were briefly centrifuged (10 s, 200 ×g) and placed on a magnet for 5 min. The clear solution containing the tryptic peptides was transferred to a fresh multiwell plate. The beads were taken up in 47 μL 25 mM ammonium bicarbonate and incubated while shaking (10 min, 1000 rpm). The plate was then placed on a magnetic stand and after 5 min the cleared supernatant was collected and combined with the recovered first peptide mix, followed by the addition of formic acid (FA) to a final concentration of 2% (v/v) for trypsin inactivation.

## Sample clean-up for LC-MS/MS

All digests were desalted on home-made C18 StageTips[65] containing two layers of an octadecyl silica membrane (CDS Analytical). All centrifugation steps were carried out at room temperature. The StageTips were first activated and equilibrated by passing 50 μL of methanol (600 ×g, 2 min), 80% (v/v) acetonitrile (ACN) with 0.5% (v/v) FA (600 ×g, 2 min) and 0.5% (v/v) FA (600 ×g, 2 min) over the tips. Next, the acidified tryptic digests were passed over the tips (800 ×g, 3 min). The immobilized peptides were then washed with 50 μL and 25 μL 0.5% (v/v) FA (800 ×g, 3 min). Bound peptides were eluted from the StageTips by application of two rounds of 25 μL 80% (v/v) ACN with 0.5% (v/v) FA (800 ×g, 2 min). Peptide samples were then dried using a vacuum concentrator (Eppendorf) and the peptides were dissolved in 15 μL 0.1% (v/v) FA prior to analysis by MS.

## Mass spectrometry

LC-MS/MS analysis of peptide samples was performed on an Orbitrap Fusion Lumos mass spectrometer (Thermo Scientific) coupled to a Vanquish Neo ultra high-performance liquid chromatography (UHPLC) system (Thermo Scientific) that was operated in the one-column mode. The analytical column was a fused silica capillary (inner diameter: 75 μm, outer diameter: 360 μm, length: 28 cm; CoAnn Technologies) with an integrated sintered frit packed in-house with Kinetex 1.7 μm XB-C18 core shell material (Phenomenex). The analytical column was encased by a PRSO-V2 column oven (Sonation) and attached to a nanospray flex ion source (Thermo Scientific). The column oven temperature was set to 50 °C during sample loading and data acquisition. The LC was equipped with two mobile phases: solvent A (2% ACN and 0.2% FA, in water) and solvent B (80% ACN and 0.2% FA, in water). All solvents were of UHPLC grade (Honeywell). Peptides were directly loaded onto the analytical column with a maximum flow rate that would not exceed the set pressure limit of 950 bar (usually around 0.5 – 0.6 μL/min) and separated on the analytical column by running a 105 min gradient of solvent A and solvent B at a flow rate of 300 nL/min (start with 3% (v/v) B, gradient 3% to 6% (v/v) B for 5 min, gradient 6% to 29% (v/v) B for 70 min, gradient 29% to 42% (v/v) B for 15 min, gradient 42% to 100% (v/v) B for 5 min and 100% (v/v) B for 10 min).

The mass spectrometer was controlled by the Orbitrap Fusion Lumos Tune Application (version 4.1.4244) and operated using the Xcalibur software (version 4.7.69.37). The MS settings are provided in Supplementary Table 1 (for measurements in data-dependent acquisition (DDA) mode; Fig. 1d, project: ACE_ACE_0869; Supplementary Fig. 4, project: ACE_0882-DDA), Supplementary Table 2 (for measurements in data-independent acquisition (DIA) mode; Fig. 5, projects: ACE_0882-DIA and ACE_0883-DIA) and Supplementary Table 3 (for measurements in DIA mode; Supplementary Fig. 5a–e, project: ACE_1029-DIA and Supplementary Fig. 5f–j, project: ACE_1030-DIA). Notably, the DDA data shown in Supplementary Fig. 4 were collected from the exact same samples as the DIA data shown in Fig. 5a–f. A direct comparison illustrates the advance the DIA method provides for a comprehensive proteome coverage.

## Data processing and analysis of DIA mass spectrometry experiments

Recorded RAW data files were converted to the mzML file format using ProteoWizard[66] ("peak Picking" (vendor MS level 1) as first filter, TPP compatibility, 64 bit encoding precision, and index writing switched on) and analyzed with DIA-NN[67] (version 1.8.1 or 2.2.0). DIA-NN was used in the library free mode. Spectral library generation is based on the Uniprot *H. sapiens* reference proteome (UP000005640_9606_OPPG; one protein per gene; 20606 entries, downloaded July 2023). For the search we selected "Trypsin/P" as protease with 2 missed cleavages allowed. As variable modifications we set "N-ter M excision", "C carbamidomethylation", "Ox(M)" and "Ac(N-term)". The maximum number of variable modifications was set to 2. Peptide length was set from 7 to 30; Precursor charge range from 2 to 6. Precursor m/z range was set to 396–1004 and Fragment ion m/z range was set to 150–1800. Mass accuracy, MS1 accuracy and Scan window was kept at "0" (automatic). Unrelated runs, use isotopologues, MBR and no shared spectra were selected. Heuristic protein inference was based on "Genes". Neural network classifier was set to "Single-pass mode". Quantification strategy was set to "Robust LC (high precision) with cross-run normalization set to "RT & signal-dep.". Library generation was set to "smart profiling" and Speed and RAM usage was set to "optimal results". Precursor FDR was set to 1%.

For further data analysis and filtering of the DIA-NN output, the report.pg_matrix.tsv file which contains the normalized (MaxLFQ)[68] protein group intensities was loaded into Perseus[69] (version 1.6.10.0). The data was transformed to the $log_2(x)$ scale and biological replicates were combined into categorical groups to allow comparison of the different treatment groups. For the full proteome analysis, only protein groups (PGs) with a valid LFQ intensity in at least two out of four replicates in each categorical group were kept for further analysis. The $log_2$-fold change in normalized protein group quantities between the different categorical groups was determined based on the mean LFQ intensities of replicate samples (relative quantification). To enable quantification, missing LFQ intensities were imputed from a normal distribution (width 0.3, down shift 1.8). The statistical significance of the difference in LFQ intensity was determined via a two-sided Student's t-test. To compare and visualize significantly up- and down-regulated (−Log(*P*-value) >2 and $Log_2$ fold change >1) proteins between both cancer entities and compound treatments of the mass spectrometry analysis, area-proportional Venn diagrams were used (https://biovenn.nl/). Pathway analysis was performed using the Metascape online platform (https://metascape.org)[70]. Graphs were visualized in R. Gene Ontology (GO) molecular functions and biological processes, canonical pathways, hallmark gene sets, Reactome gene sets, and Kyoto Encyclopedia of Genes and Genomes (KEGG) pathways were chosen as reference datasets to identify correlated biological processes among the tested proteins.

## Data processing and analysis of DDA mass spectrometry experiments

RAW spectra were submitted to an Andromeda[71] search in MaxQuant (version 2.5.2.0) using the default settings[72]. Label-free quantification and match-between-runs was activated[68]. The MS/MS spectra data were searched against the Uniprot *H. sapiens* reference proteome (see above). All searches included a contaminants database search as implemented in MaxQuant with 245 entries. Andromeda searches allowed oxidation of methionine residues (16 Da) and acetylation of the protein N-terminus (42 Da). Carbamidomethylation on Cystein (57 Da) was selected as static modification. Enzyme specificity was set to "Trypsin/P". The instrument type in Andromeda searches was set to Orbitrap and the precursor mass tolerance was set to ±20 ppm (first search) and ±4.5 ppm (main search). The MS/MS match tolerance was set to ±0.5 Da. The peptide spectrum match FDR and the protein FDR were set to 0.01 (based on target-decoy approach). For protein quantification unique and razor peptides were allowed. Modified peptides were allowed for quantification. The minimum score for modified peptides was 40. Label-free protein quantification was switched on, and unique and razor peptides were considered for quantification with a minimum ratio count of 2. Retention times were recalibrated based on the built-in nonlinear time-rescaling algorithm. MS/MS identifications were transferred between LC-MS/MS runs with the "match between runs" option in which the maximal match time window was set to 0.7 min and the alignment time window set to 20 min. The quantification is based on the "value at maximum" of the extracted ion current. At least two quantitation events were required for a quantifiable protein. Further analysis and filtering of the results was done in Perseus[69] v1.6.10.0 as described above. Comparison of protein group quantities (relative quantification) between different runs is based solely on LFQ values as calculated by the MaxQuant MaxLFQ algorithm[68].

## Statistics and reproducibility

All results were consistently observed in at least two, typically three independent experiments. In addition, western blots assessing PRO-TAC degradation efficiencies were performed in three independent laboratories, further confirming reproducibility.

## Reporting summary

Further information on research design is available in the Nature Portfolio Reporting Summary linked to this article.

## Data availability

Mass spectrometry raw data generated in this study have been deposited to the ProteomeXchange Consortium through the PRIDE partner repository under accession codes PXD063889, PXD063917, PXD063920, PXD063913 and PXD070431. 3′ mRNA-sequencing data generated in this study have been deposited to the Biostudies database under accession number E-MTAB-16302. Processed mass spectrometry data are enclosed as Supplementary Data 1 (for Fig. 1d), Supplementary Data 2 (for Fig. 5a-f), Supplementary Data 3 (for Fig. 5g-l), Supplementary Data 4 (for Supplementary Fig. 4) Supplementary Data 5 (for Supplementary Figs. 5a-e) and Supplementary Data 6 (for Supplementary Figs. 5f-j). Uncropped blots are provided as Source Data. Chemical compound characterization data are provided in the Supplementary Information. The *H. sapiens* reference proteome used in this study is available in the Uniprot database under accession code UP000005640_9606_OPPG. The RNA-seq data of the MGA knock-out used in this study is available in the BioStudies database under accession code E-MTAB-6005. The crystal structure of USP7 in complex with FT671 used in this study is available in the pdb database under accession code 5NGE. Source data are provided with this paper.

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

## Acknowledgements

We are grateful to all members of the Paschen, Grüner and Gersch labs for discussions, advice, and reagents. We thank Mirko Schmidt for initial work on USP7 PROTACs, and Zhou Zhao for the synthesis of fluorescent ubiquitin probes. We thank Jenny Borman, Finn Pfeffermann, Eva Hahn, Elisa Krosta and Nancy Meyer as well as the NMR and HRMS facilities of TU Dortmund University headed by Bastian Grabe and Sebastian Zühlke for excellent technical assistance. We are grateful to G. Peters for the gift of a USP7 plasmid (Addgene # 46753). This work was funded by Deutsche Forschungsgemeinschaft (DFG, German Research Foundation, Project-ID 424228829 – SFB1430, to F.K., M.K., N.S., S.K., A.P., B.M.G, M.G.). Work in the Gersch lab is further supported by AstraZeneca, Merck KGaA, Pfizer Inc., and the Max Planck Society as part of the Chemical Genomics Center III (CGCIII-352S, to M.G.), and by the DFG through an Emmy Noether Award (GE 3110/1-1 to M.G.). The Grüner lab is further supported by the DFG with an Emmy Noether Award (GR4575/1-1, GR4575/1-2, to B.M.G). Work in the Paschen lab was additionally funded by the DFG (Project-ID 418179183 KFO 337; PA 2376/1-2, to. A.P.). Work in the Wolf lab was supported by grants from the German Research Foundation (DFG: WO 2108/2-1, TRR387, GRK 3085), the German Cancer Aid (DKH: TACTIC) and the European Research Council (ERC: PROTAC-PDAC: #101087045) to E.W. Work in the Knapp lab was further supported by the Structural Genomics Consortium (SGC, a registered charity (no: 1097737) that receives funds from Bayer AG, Boehringer Ingelheim, Bristol Myers Squibb, Genome Canada), by the EU/EFPIA/OICR/McGill/KTH/Diamond Innovative Medicines Initiative 2 Joint Undertaking (EUbOPEN grant 875510), Janssen, Pfizer and Takeda, as well as by the LOEWE Center Frankfurt Cancer Institute (FCI) funded by the Hessian Ministry of Higher Education, Research and the Arts (III L5-519/03/03.001-(0015)), all to S.K.

## Author contributions

N.K., S.U., J.A.S., A.P., B.M.G., and M.G. jointly designed the study, planned experiments and analyzed data. N.K. performed chemical synthesis as well as protein inhibition and degradation assays. S.U., M.D., J.J., J.A.S., and P.S. performed cellular experiments with PDAC and melanoma cells, respectively. B.A. with support by J.M. and under supervision by E.W. generated HiBiT-USP7 cells and recorded HiBiT data. M.V. analyzed RNA-seq data. M.P.S. under supervision by S.K. recorded ternary complex formation data. F.K. and M.K. performed mass spectrometry. S.F. performed the pulldown assay. J.K. and N.S. performed cell imaging. N.K., S.U., J.A.S., A.P., B.M.G., and M.G. together assembled the manuscript with input from all authors.

## Funding

## Competing interests

The authors declare no competing interests.

## Additional information

[1]Department of Chemistry and Chemical Biology, TU Dortmund University, Dortmund, Germany. [2]Max Planck Institute of Molecular Physiology, Dortmund, Germany. [3]West German Cancer Center, Department of Medical Oncology, University Hospital Essen, University of Duisburg-Essen, Essen, Germany. [4]Department of Dermatology, University Hospital Essen, University of Duisburg-Essen, Essen, Germany. [5]Institute of Biochemistry, University of Kiel, Kiel, Germany. [6]Institute of Pharmaceutical Chemistry, Goethe University Frankfurt, Frankfurt am Main, Germany. [7]Structural Genomics Consortium (SGC), Buchmann Institute for Molecular Life Sciences (BMLS), Frankfurt am Main, Germany. [8]German Cancer Consortium, DKTK partner site Essen/Düsseldorf, a partnership between DKFZ and University Hospital Essen, University Duisburg-Essen, Essen, Germany. [9]Faculty of Biology and Analytics Core Facility Essen (ACE), University of Duisburg-Essen, Essen, Germany. [10]Imaging Center Campus Essen, Center of Medical Biotechnology, University of Duisburg-Essen, Essen, Germany. [11]Faculty of Biology and Center of Medical Biotechnology, University of Duisburg-Essen, Essen, Germany. [12]German Cancer Consortium, DKTK partner site Frankfurt-Mainz, German Cancer Research Center (DKFZ), Heidelberg, Germany. [13]These authors contributed equally: Nikolas Klink, Sebastian Urban, Johanna A. Seier. ✉e-mail: annette.paschen@uk-essen.de; barbara.gruener@uk-essen.de; malte.gersch@tu-dortmund.de

