## [Transparent Peer Review file · Nature Communications]

Targeted degradation of USP7 in solid cancer cells reveals distinct effects of deubiquitinase degraders and inhibitors

Corresponding Author: Dr Malte Gersch

Version 0:

Reviewer comments:

Reviewer #1

(Remarks to the Author)

In this outstanding contribution, Klink et al. provide a direct comparison between USP7 inhibition and targeted degradation in cancer cells. USP7 has long been pursued as a therapeutic target in oncology, with several small-molecule inhibitors advancing into preclinical development. However, because USP7 broadly regulates the ubiquitin–proteasome system, its functions are highly context-dependent across cell types. Defining these context-specific roles—as well as the contribution of non-catalytic functions—is essential for the rational development of USP7-targeted therapies. Targeted protein degraders offer a powerful means to dissect catalytic versus non-catalytic contributions to USP7 function.

By combining structural elements from two potent USP7 inhibitors—FT671 and Compound 5—the authors developed NK192, which maintains low nanomolar potency while incorporating a more accessible exit vector for linker attachment and VHL recruitment. Subsequent optimization yielded two effective degraders, NK250 and NK266, with degradation activity comparable to the parent inhibitor. While previous studies have converted USP7 inhibitor scaffolds into degraders, this work distinguishes itself by directly comparing the cellular consequences of inhibition and degradation. The authors also rigorously characterize the degraders in terms of ternary complex formation and efficacy.

A particularly compelling finding is that USP7 degraders offer greater selectivity than inhibitors for probing USP7 function. The authors support this conclusion with experiments demonstrating that inhibitors—but not degraders—promote increased glucose consumption. Overall, this study introduces well-characterized inhibitor/degrader pairs and underscores the value of such tools in dissecting the multifaceted roles of USP7. The work is rigorous, and the findings are both novel and impactful. I recommend publication without revision.

Reviewer #2

(Remarks to the Author)

This manuscript presents a technically rigorous and comprehensive chemical biology investigation comparing USP7 inhibition and targeted degradation via PROTACs in solid tumour models (PDAC and melanoma). The authors successfully develop a matched chemical pair, a selective inhibitor (NK192) and degraders (NK250/NK266), and demonstrate their differential effects using proteomic, biochemical, and phenotypic assays. The study provides comparative proteomic profiling of inhibition vs. degradation, demonstrating modality-specific effects with broader changes in protein expression upon inhibition and more selective responses upon degradation. The degraders showed cell-type–specific degradation kinetics, highlighting differences in proteomic remodelling between PDAC (Panc89) and melanoma (Ma-Mel-47) cell lines. Metabolic phenotypes, including glucose sensitivity, OCR shifts, and media acidification, were observed, particularly under inhibitor treatment, indicating functional consequences of USP7 modulation.

The reviewers commend the strength of the chemical tools, the specificity of the PROTACs, and the systematic proteomic analyses. However, they raise critical concerns regarding compound dose inconsistency in proteomics, lack of functional validation, and the unclear rationale behind certain phenotypic assays (metabolism in Figure 6).

Major Issues

- 1. Dose Inconsistency in Proteomics (Figure 5):** The PROTACs (NK250/NK266) were used at 1 μ M, while the inhibitor (NK192) was used at 5 μ M, despite their comparable DC50 and IC50 values. This disparity undermines the comparative conclusion that inhibition causes broader proteomic changes than degradation. In Figure 5, DIA results show that 5 μ M NK192 (72 h) induces significant changes in protein expression, whereas 1 μ M NK250/NK266 induces fewer. These differences may reflect secondary or off-target effects of the inhibitor at a higher concentration. Therefore, it is difficult to conclude that inhibition causes broader proteome-wide effects than degradation. The authors should (1) re-perform the proteomics experiments using dose-matched conditions, or, if different concentrations are essential, provide a clear rationale in the text, and (2) validate key USP7 substrates (e.g., PRC1.6, TRIP12) using Western blot under matched conditions to distinguish modality and kinetic effects.
- 2. DDA vs. DIA Proteomics Concerns (Figure 5 and SI):** The manuscript includes both DDA and DIA datasets; however, DDA is limited to Panc89 and offers lower protein coverage, while DIA includes both Panc89 and Ma-Mel-47 and provides broader data. Including both datasets without justification may confuse readers or dilute the conclusions. As DIA sufficiently supports the main findings, the reviewer recommends focusing on the DIA dataset in the main text. If both methods are necessary, the authors should clearly explain the rationale and how the datasets complement each other.
- 3. Biological Relevance and Functional Validation:** The biological consequences of proteomic changes (e.g., PRC1.6, E3 ligase targets) are not explored in depth. It remains unclear whether transcription, chromatin states, or downstream signalling pathways are affected. Moreover, there is no validation of known or novel USP7 substrates beyond USP7 itself. The study would benefit from validation (Western blot, qPCR) of key hits to link protein changes to functional outcomes.
- 4. Metabolic Phenotyping (Figure 6):** The metabolic readouts (media acidification, apoptosis, OCR, glucose dependence) appear disconnected from USP7 function and the upstream proteomic data. The authors should provide a stronger rationale or mechanistic link between USP7 biology and the observed metabolic effects. If this cannot be established, they should consider adding or replacing with phenotypes more directly linked to USP7 function (e.g., DNA damage response, cell cycle, chromatin remodelling).
- 5. Cell-Type Specificity:** The differences in degradation efficiency and proteomic remodelling between Panc89 and Ma-Mel-47 are not been thoroughly analysed. GO/pathway enrichment and clustering analyses could help elucidate the mechanistic basis of these differences.
- 6. Genetic Validation:** The study relies solely on chemical perturbation. No orthogonal validation (e.g., CRISPR knockout, siRNA knockdown) is provided to confirm on-target effects. Including at least one genetic perturbation experiment (USP7 knockout or knockdown) would strengthen mechanistic claims and help rule out off-target effects.

Minor Issues

- The IC50 value for NK264 (15 nM) should be included below its chemical structure. The IC50 curve could also be added to Figure 1C for clarity.
- In Figure 1D, the x-axis should include concentration, and the roles of NK192 (as competitor) and NK264 (as probe) should be clearly explained.
- In Figure 3, the authors state that NK266 is more effective than NK250 in Ma-Mel-47 cells. However, this is not convincingly demonstrated. The authors should show DC50 values for both degraders in both cell lines with statistical analysis. They should also explain why NK266 performs better in Ma-Mel-47 despite targeting the same protein.
- In Figure 3C, DC50 values should be quantified as done in Supporting Information Figures 2a and 2b, for consistency.
- Figures 3b–e, 3g, 4a–c, and 6b–e should include complete experimental details, such as the cell line, compound concentration, and incubation time, to improve interpretability.
- In Figure 6, the media colour changes shown in panel 6a should be quantified and reported for both cell lines. Additionally, graphs in panels 6b–e could be combined side-by-side to improve visual clarity.
- In the introduction, the authors should clarify that although USP7 PROTACs have previously been reported (Pei et al., *Angew Chem Int Ed*, 2022; DOI: 10.1002/anie.202204395), this work provides the first matched inhibitor–degrader pair specifically developed and characterised in solid tumour models.

Reviewer #3

(Remarks to the Author)
see attached review

Version 1:

Reviewer comments:

Reviewer #2

(Remarks to the Author)

The revised manuscript has been substantially improved and fully addresses all major and minor concerns raised in the initial review. The authors have responded thoroughly and convincingly, supported by extensive new experimental data that significantly strengthen the rigour and mechanistic depth of the study.

The previously noted dose inconsistency in the proteomics experiments has been comprehensively resolved through dose-matched analyses and supporting validation experiments, which confirm that the broader proteomic changes observed upon

USP7 inhibition are modality-specific rather than concentration-driven. The rationale for prioritising DIA proteomics is now clear, and the inclusion of DDA data as supporting evidence is appropriate.

The authors have also added strong biological and functional validation, including pathway enrichment, transcriptomic analyses, and genetic USP7 knockout studies, which collectively provide compelling mechanistic insight into the consequences of USP7 degradation versus inhibition. The metabolic phenotyping is now clearly framed as an inhibitor-associated, USP7-independent effect, further strengthening the conceptual clarity of the work.

All minor issues related to data presentation and clarity have been addressed. The only remaining minor issue is that the label of one compound, which I think is "NK250", appears to be missing in Supporting Figure 7d and should be added.

Overall, the manuscript now represents a high-quality and well-substantiated contribution to the field. I recommend publication in its current form, pending this minor correction.

Reviewer #3

(Remarks to the Author)

This reviewer commends the authors for their thorough and complete responses to the previous comments for the manuscript received from all of the reviewers. The inclusion of the additional data and clarification points and context of the positions taken in presenting their work have significantly improved the manuscript. As a result of the additional data and edits this reviewer would recommend this be published without further revision [editorial note: confidential, unpublished data redacted]

We would like to sincerely thank all reviewers for their careful analysis of our work and for their constructive suggestions. We were truly thrilled to read their supportive comments. We have addressed all raised points in full through additional analyses and experiments, and we hope that the manuscript is now judged ready for publication.

Reviewer #1 (Remarks to the Author)

In this outstanding contribution, Klink et al. provide a direct comparison between USP7 inhibition and targeted degradation in cancer cells. USP7 has long been pursued as a therapeutic target in oncology, with several small-molecule inhibitors advancing into preclinical development. However, because USP7 broadly regulates the ubiquitin–proteasome system, its functions are highly context-dependent across cell types. Defining these context-specific roles—as well as the contribution of non-catalytic functions—is essential for the rational development of USP7-targeted therapies. Targeted protein degraders offer a powerful means to dissect catalytic versus non-catalytic contributions to USP7 function.

By combining structural elements from two potent USP7 inhibitors—FT671 and Compound 5—the authors developed NK192, which maintains low nanomolar potency while incorporating a more accessible exit vector for linker attachment and VHL recruitment. Subsequent optimization yielded two effective degraders, NK250 and NK266, with degradation activity comparable to the parent inhibitor. While previous studies have converted USP7 inhibitor scaffolds into degraders, this work distinguishes itself by directly comparing the cellular consequences of inhibition and degradation. The authors also rigorously characterize the degraders in terms of ternary complex formation and efficacy.

A particularly compelling finding is that USP7 degraders offer greater selectivity than inhibitors for probing USP7 function. The authors support this conclusion with experiments demonstrating that inhibitors—but not degraders—promote increased glucose consumption. Overall, this study introduces well-characterized inhibitor/degrader pairs and underscores the value of such tools in dissecting the multifaceted roles of USP7. The work is rigorous, and the findings are both novel and impactful. I recommend publication without revision.

We are very grateful to this reviewer for the appreciation of our work, the careful analysis and the supportive conclusion.

Reviewer #2 (Remarks to the Author)

This manuscript presents a technically rigorous and comprehensive chemical biology investigation comparing USP7 inhibition and targeted degradation via PROTACs in solid tumour models (PDAC and melanoma). The authors successfully develop a matched chemical pair, a selective inhibitor (NK192) and degraders (NK250/NK266), and demonstrate their differential effects using proteomic, biochemical, and phenotypic assays. The study provides comparative proteomic profiling of inhibition vs. degradation, demonstrating modality-specific effects with broader changes in protein expression upon inhibition and more selective responses upon degradation. The degraders showed cell-type–specific degradation kinetics, highlighting differences in proteomic remodelling between PDAC (Panc89) and melanoma (Ma-Mel-47) cell lines. Metabolic phenotypes, including glucose sensitivity, OCR shifts, and media acidification, were observed, particularly under inhibitor treatment, indicating functional consequences of USP7 modulation.

The reviewers commend the strength of the chemical tools, the specificity of the PROTACs, and the systematic proteomic analyses. However, they raise critical concerns regarding compound dose inconsistency in proteomics, lack of functional validation, and the unclear rationale behind certain phenotypic assays (metabolism in Figure 6).

We thank these reviewers for their encouraging comments, recognizing the value of our chemical toolbox and selective USP7 degraders. We acknowledge the raised concerns, which we have fully addressed with our revised and strengthened manuscript as described below.

Major Issues

1. Dose Inconsistency in Proteomics (Figure 5): The PROTACs (NK250/NK266) were used at 1 μ M, while the inhibitor (NK192) was used at 5 μ M, despite their comparable DC50 and IC50 values. This disparity undermines the comparative conclusion that inhibition causes broader proteomic changes than degradation. In Figure 5, DIA results show that 5 μ M NK192 (72 h) induces significant changes in protein expression, whereas 1 μ M NK250/NK266 induces fewer. These differences may reflect secondary or off-target effects of the inhibitor at a higher concentration. Therefore, it is difficult to conclude that inhibition causes broader proteome-wide effects than degradation.

The authors should (1) re-perform the proteomics experiments using dose-matched conditions, or, if different concentrations are essential, provide a clear rationale in the text, and (2) validate key USP7 substrates (e.g., PRC1.6, TRIP12) using Western blot under matched conditions to distinguish modality and kinetic effects.

We agree with this comment and have addressed it through both suggested options:

- (1) We recorded additional data by re-performing the proteomics experiments in both cell lines using the inhibitor NK192 and the PROTACs NK250 and NK266 at dose-matched concentrations. All experiments are now available at 1 μ M and at 5 μ M for the 72 h timepoint (see new Supporting Fig. 5 together with Fig. 5). Under dose-matched conditions, we observed the exact same effects as in our previous experiments. Importantly, treatment with only 1 μ M inhibitor NK192 still induced significant proteomic changes (Supporting Fig. 5c and h). In contrast, treatment with PROTACs at 5 μ M retained exquisite specificity for USP7 depletion, accompanied by selective downregulation of USP7 substrates (Supporting Fig. 5b and g). This is consistent with the metabolic rewiring already occurring with 1 μ M inhibitor, which does not occur even with 10 μ M PROTAC (Fig. 7). We concur with the reviewer that some of these differences observed upon longer treatment times (72 h and beyond) likely stem from non-USP7-related effects by the inhibitor. We have substantiated this with additional experiments (new elements in Fig. 7) as described below. Moreover, we have strengthened our conclusions through the incorporation of negative control compound NK245 (structure similar to NK250, incapable of engaging the VHL ligase) into the new proteomics data (Supporting Fig. 5e and j).
- (2) In addition, we validated the specific effects mediated by our degraders. We confirmed the downregulation of selected USP7 substrates by Western blot analysis. As suggested by the reviewer, all experiments were carried out under matched concentration conditions (1 μ M and 5 μ M, see new Fig. 6a+b). These data support the conclusion that the observed differences between inhibition and degradation are not driven by concentration-dependent effects.

2. DDA vs. DIA Proteomics Concerns (Figure 5 and SI): The manuscript includes both DDA and DIA datasets; however, DDA is limited to Panc89 and offers lower protein coverage, while DIA includes both Panc89 and Ma-Mel-47 and provides broader data. Including both datasets without justification may confuse readers or dilute the conclusions. As DIA sufficiently supports the main findings, the reviewer recommends focusing on the DIA dataset in the main text. If both methods are necessary, the authors should clearly explain the rationale and how the datasets complement each other.

We thank the reviewers for this comment and agree that the DIA dataset provides a broader proteome coverage and sufficiently supports the main conclusions of the study. Accordingly, the DIA data are the primary basis for the proteomics-related conclusions and analyses in the main figures. However, we chose to keep the DDA dataset in the Supporting Information to substantiate the proteomic changes observed in the DIA dataset. These consistent data support the robustness of our analyses. Moreover, we find this clean comparison of data recorded from the exact same samples (where DIA led to almost twice the number of quantified protein groups compared to DDA) to be instructive for other groups. We have explained this more clearly in the manuscript as requested. We hope that reviewers support the strategic decision to keep both datasets in the manuscript.

3. Biological Relevance and Functional Validation: The biological consequences of proteomic changes (e.g., PRC1.6, E3 ligase targets) are not explored in depth. It remains unclear whether transcription,

chromatin states, or downstream signalling pathways are affected. Moreover, there is no validation of known or novel USP7 substrates beyond USP7 itself. The study would benefit from validation (Western blot, qPCR) of key hits to link protein changes to functional outcomes.

To address the reviewers' concerns regarding the biological relevance and functional consequences of the observed proteomic changes, we expanded our analyses to directly link USP7 degradation to chromatin-associated processes by characterizing its downstream effects on transcription.

- Firstly, we added pathway enrichment analysis of differentially abundant proteins upon USP7 PROTAC treatment, validating our key finding of PRC1.6 downregulation across both entities (new Fig. 6d+e). This suggested an upregulation of normally PRC-repressed genes.
- Secondly, to address the question of downstream regulation, we performed a 3' bulk mRNA sequencing analysis of PROTAC-treated Ma-Mel-47 and Panc89 cells. This enabled the comprehensive assessment of the consequences of USP7 degradation, in particular at the transcriptome level, through changes in PRC1.6 component abundance. To this end, we compared our transcriptome data to published MGA (a PRC1.6 component) knock out transcriptome data and observed a very strong correlation (see new Fig. 6f+g; this is despite the differences in experimental setup, duration of perturbation and cell line). This highlights how USP7 degradation elicits MGA-dependent functional consequences, thereby supporting the striking effects on chromatin upon USP7 degradation (Fig. 6).
- Thirdly, using dose-matched concentrations of inhibitor and PROTAC, we validated our transcriptomic approach via qPCR in Panc89 cells, focusing on two of the most upregulated genes in both the MGA-KO and USP7-PROTAC dataset (*SYCE2*, *SMC1B*, shown on the right). In agreement with the MGA-KO dataset, both inhibition and degradation of USP7 resulted in the transcriptional upregulation of both genes. As we consider the mRNA sequencing data to be comprehensive, we did not include the qPCR data in our manuscript.

Regarding the requested validation of USP7 substrates by Western Blot, we included data in Fig. 6a+b on TRIP12 and TRIM27 depletion upon USP7 modulation.

4. Metabolic Phenotyping (Figure 6): The metabolic readouts (media acidification, apoptosis, OCR, glucose dependence) appear disconnected from USP7 function and the upstream proteomic data. The authors should provide a stronger rationale or mechanistic link between USP7 biology and the observed metabolic effects. If this cannot be established, they should consider adding or replacing with phenotypes more directly linked to USP7 function (e.g., DNA damage response, cell cycle, chromatin remodelling).

We thank the reviewers for this comment and, as suggested by the reviewers, have added phenotypic data directly linked to USP7 function on transcription (see our response above for point 3). However, we have also better explained why the data on the metabolic phenotype (new Fig. 7) are included. The effects the reviewer mentioned (media acidification, apoptosis under low glucose conditions, etc) were consistently observed when cells were treated with USP7 inhibitors NK192 or FT671, but not with PROTACs. The effects were especially pronounced upon longer treatment times (e.g. at 72 h). In the preceding figures (Figs. 5+6), we explored the specific effects of USP7 degradation and observed large scale proteomic changes upon inhibitor treatment. This suggested also drastic phenotypic changes (new Fig. 6h-i). Indeed, we observed the described effects with the inhibitors but not with PROTAC treatment. This represents a phenotypic distinction that parallels the differences in proteomic responses between these two modalities.

We present these data as an example of how PROTAC treatment enables the investigation of on-target USP7 biology with longer treatment times. More broadly, they highlight the possibility to achieve elevated target specificity of a degrader compared to an inhibitor. This distinction is substantiated by our inhibitor-treated USP7-KO data (Fig. 7, see our response to point 6 below), supporting the value of our chemical toolbox for this purpose.

5. Cell-Type Specificity: The differences in degradation efficiency and proteomic remodelling between Panc89 and Ma-Mel-47 are not been thoroughly analysed. GO/pathway enrichment and clustering analyses could help elucidate the mechanistic basis of these differences.

In the revised version of our manuscript, we included comprehensive pathway analyses of differently regulated proteins in both cell lines (new Fig. 6d+e and new Supporting Fig. 6a-d). These data highlight the selective regulation of PRC-related processes by our degraders, as well as cell-line-specific effects observed across all replicates. Importantly, as PROTAC NK250 could not sufficiently degrade USP7 in Ma-Mel-47 cells (see Supporting Fig. 2), we used cell line optimized PROTACS (NK250 for PDAC and NK266 for melanoma). These are able to fully deplete USP7 in the respective cell lines (near 100 % D_{max} , see new Fig. 3d-g for degradation efficiency quantifications as well as Fig. 7c and Supporting Fig. 7d for 72 h treatment data). In both cell lines, PRC- and ubiquitination-associated pathways were found as most downregulated processes upon PROTAC treatment with high significance (Fig. 6d-e). As suggested by the reviewers, we now highlight similarities and also differences across both cancer types (see Fig. 6h).

6. Genetic Validation: The study relies solely on chemical perturbation. No orthogonal validation (e.g., CRISPR knockout, siRNA knockdown) is provided to confirm on-target effects. Including at least one genetic perturbation experiment (USP7 knockout or knockdown) would strengthen mechanistic claims and help rule out off-target effects.

We appreciate the reviewers for this valuable suggestion and have added genetic validation to our key claims. To provide genetic perturbation and thereby confirm the on-target effects of our USP7 degraders, we attempted the knockout of USP7 in PDAC cell lines Panc89 and AsPC1 as well as in the melanoma cell line Ma-Mel-47. Despite extensive attempts, we were unable to obtain a USP7 knockout clone in any PDAC cell line (see below left Fig. panels a-b, right Fig. panel a). While some cells showed reduced USP7 levels, they frequently also developed large vacuole-like bubbles (right Fig. b); a phenotype commonly observed upon gene knockout in these cell lines but not observed upon complete USP7 depletion by PROTACs.

The difficulty to establish a complete USP7 knockout in PDAC cells thus highlights the critical need for chemical tools, as provided in our manuscript, that enable the investigation of USP7-mediated phenotypes and are broadly applicable to different cell lines.

Importantly, we successfully generated a Ma-Mel-47 USP7-KO by single-cell cloning. Using this cell line, we confirmed the downregulation of the identified USP7 substrates TRIP12, TRIM27, and PCGF6 in USP7-deficient cells (new Fig. 6c). These data fulfill the reviewers' request for a genetic validation of our key claim of the PROTAC's on-target activity.

Furthermore, this available USP7 KO line allowed us to convincingly demonstrate that the metabolic perturbations after extended inhibitor treatment (72 h), characterized by increased glucose dependency, impaired oxygen consumption and medium acidification, are independent of USP7 (new Fig. 7f+g+h). These data thus highlight an important caveat when hydroxypiperidine-based USP7 inhibitors such as NK192 or FT671 are used with longer treatment times. These findings are summarized in a scheme in Fig. 7i. They imply that structurally similar USP7 inhibitors are not entirely specific, yet the identification of other cellular targets through a pull-down experiment out of cell lysate (Fig. 1d) did not lead to candidate proteins. Our mass spectrometry experiments revealed that the proteomic changes observed within 6 h to 24 h of USP7 inhibitor treatment are very similar to those observed upon USP7 degradation after 24 h. This stresses that the use of structurally similar USP7 inhibitors for short treatment times appears suitable, yet data generated from longer treatment times (particularly 72 h and beyond) or under glucose-deprived conditions should be interpreted with care. Concurrently, the data highlight the increased specificity of our degraders and the high utility of selective tools for the investigation of USP7 in cells.

To validate this claim in an additional cell line, we also generated a USP7 KO in the melanoma cell line A375 (shown on the right). Consistent with our findings in Ma-Mel-47 cells, USP7 knockout in A375 cells resulted in non-specific metabolic rewiring upon prolonged inhibitor treatment (evident from rapid media acidification upon NK192 treatment). To keep the manuscript streamlined and focused on the cell lines Panc89 and Ma-Mel-47, we have decided to not include this data in the final manuscript.

Minor Issues

1. The IC50 value for NK264 (15 nM) should be included below its chemical structure. The IC50 curve could also be added to Figure 1C for clarity.

Thank you, we have changed the figure as suggested in both ways to improve clarity.

2. In Figure 1D, the x-axis should include concentration, and the roles of NK192 (as competitor) and NK264 (as probe) should be clearly explained.

We added clarifying text to Fig. 1d to better convey the roles of both compounds in this experiment. The experiment was carried out with a fixed concentration of NK192 which is stated in the caption (confirming that the x-axis is the log2-fold change of protein amounts detected in the DMSO vs. the NK192-treated samples). Moreover, we also added the descriptor "biotin probe" to the chemical structure of NK264 in Fig. 1b.

3. In Figure 3, the authors state that NK266 is more effective than NK250 in Ma-Mel-47 cells. However, this is not convincingly demonstrated. The authors should show DC₅₀ values for both degraders in both cell lines with statistical analysis. They should also explain why NK266 performs better in Ma-Mel-47 despite targeting the same protein.

We thank the reviewers for this suggestion. We have performed new gel-based degradation assays for both NK266 and NK250, in both cell lines and in biological triplicates. Densitometry-based DC₅₀ values are now included as new Figs. 3f+g and in Supporting Fig. 2 (raw data for all replicates are provided in the Supporting Information). These data confirm that in Ma-Mel-47 cells NK266 (DC₅₀: 29 ± 8 nM) is more effective at depleting USP7 than NK250 (DC₅₀: 154 ± 38 nM). These data substantiate our choice to proceed with the application of NK250 in Panc89 cells (DC: 8 ± 1 nM), and of NK266 in Ma-Mel-47. We currently cannot provide a rigorous explanation for this observation. While many effects could contribute, we speculate that differences in transporter protein expression or substrate recognition could be a reason for PROTAC sensitivity differences, as was recently described by Wolf et al. *Cell Chem Biol* 2025, doi: 10.1016/j.chembiol.2024.11.009.

4. In Figure 3C, DC₅₀ values should be quantified as done in Supporting Information Figures 2a and 2b, for consistency.

We have performed the required additional experiments and now provide DC₅₀ values as described in the response above. Of note, these assays were carried out in a gel-based format and thus allowed the analysis of endogenous USP7 degradation in both cell lines. They thus complement the degradation quantification in MV4-11 cells stably expressing HiBiT-tagged USP7 (in Fig. 4).

5. Figures 3b–e, 3g, 4a–c, and 6b–e should include complete experimental details, such as the cell line, compound concentration, and incubation time, to improve interpretability.

To improve clarity, we re-arranged the panels in Fig. 3 and Fig. 7 (formerly 6). Moreover, we added the corresponding cell line names to Fig. 3 and Fig. 7. In addition, we ensured that the captions of Figures 3, 4 and 7 figures include complete experimental details (as suggested: cell line, compound concentration and incubation time) for all panels. We are convinced that these changes in the figures and captions fully address these reviewers' suggestion, and that they enhance data interpretability.

6. In Figure 6, the media colour changes shown in panel 6a should be quantified and reported for both cell lines. Additionally, graphs in panels 6b–e could be combined side-by-side to improve visual clarity.

Thank you, we have quantified the media colour changes for both Panc89 and Ma-Mel-47 cell lines by measuring absorbance at 570 nm (corresponding to the pH-sensitive absorbance maximum of phenol red, the coloured component in cell culture media). These new data are provided as Fig. 7b and Supporting Fig. 7c. They enhance the visual clarity of the effects shown in the media dish images. We have also adopted the suggestion to combine the apoptosis assay data (formerly three graphs in Fig. 6c) into one panel (now Fig. 7e).

7. In the introduction, the authors should clarify that although USP7 PROTACs have previously been reported (Pei et al., *Angew Chem Int Ed*, 2022; DOI: 10.1002/anie.202204395), this work provides the first matched inhibitor–degrader pair specifically developed and characterised in solid tumour models.

We thank the reviewers for acknowledging the novelty of our characterized chemical toolbox in solid tumours models. As suggested, we have clarified the context in which we reference the Pei et al. manuscript as well as other DUB degrader manuscripts in our introduction (added text in italics: “Notably, while the targeted degradation of DUB proteins has been explored (*including degraders for USP7*), (refs 21-26, incl. Pei et al.) studies describing matched pairs of well characterized inhibitors and PROTACs do not exist for DUBs.” Moreover, we have strengthened the introduction by stating that this work indeed provides the first matched inhibitor-degrader pair. Importantly, characterising the effects of USP7 depletion in solid tumour cells (where it is non-toxic, and thus USP7-dependent proteomic changes can be studied) also sets our approach apart from previous studies which focused on AML cell lines where USP7 inhibition and degradation are toxic.

Reviewer #3 (Remarks to the Author):

The article describes the synthesis USP7 VHL derived PROTAC array and thereafter characterization of the commensurate degradation of USP7 in 2 cell lines (melanoma and pancreatic ductal) by the compounds synthesized in this array. The manuscript also describes a BRET assay providing insight into ternary complex formation along with proteomic data associated with the treatment of those cells with an optimal ligand from the array. In contrast to these data, treatments with USP7 inhibitors uncovered contrasting effects at the cellular level including metabolic changes induced by prolonged (USP7 inhibitor) treatment. Overall, the authors claim to have provided a toolbox of comprehensive reagents resulting from this work, and more generally a strategy to distinguish the catalytic activities of DUBs from their non-catalytic functions.

The manuscript was well written.

This reviewer would support publication in Nature Communications, with minor revision following the consideration of the following points:

We are very grateful for this reviewer's recommendation of publication and strong support of our work.

This reviewer would ask the authors

1. to strengthen the chemistry view of the VHL-degrader "library" and its design: specifically,
 1. Could they comment on and describe the design elements that drove the selection of the derivatives made in the "second generation" library: this reviewer would consider 15 compounds covering the chemical space is a good starting point but not comprehensive
 2. Were other vectors and chemistries considered, from the VHL ligand itself, beyond the N-substituent that they exemplified

We thank the reviewer for the appreciation of the presented chemistry.

1. We agree that 15 PROTACs are not a comprehensive collection of degraders. While we did synthesize a total of 36 USP7 PROTACs as part of this project, we chose to keep the shown SAR minimal to improve the visual and logical clarity in our manuscript. Our initial PROTACs targeting USP7 were synthesized featuring both the XL188 scaffold (Lamberto et al. *Cell Chem Biol* 2017, doi: 10.1016/j.chembiol.2017.09.003) (shown below, upper part) and a methyl-cyclopropyl-containing derivative of NK192 (shown below, lower part, similar to the later published ADC-159, Jurisic et al. *Clin Transl Med.* 2024, doi: 10.1002/ctm2.1648). All these initial molecules featured either long alkyl or PEG linkers (see Figure on the next page).

However, these did not yield any potent USP7 degraders in all cell lines tested. Due to the different scaffolds (XL188, ADC-159, NK192), we did not include these compounds in the manuscript; instead, we decided to reduce complexity by only keeping NK192 as the base scaffold. Regarding design elements: Since linear linkers did not work well in our hands, we focused on mostly rigid linkers in the degraders featured in the manuscript. Both exit vectors of USP7- and VHL-ligands were structurally defined by available co-crystal structures, which allowed a rational design of rigid linkers that would be suitable for a focused linker library as highlighted in the manuscript.

2. Indeed, we had considered utilizing the phenolic exit vector that is frequently explored in VHL ligands; however, we were also intrigued by recent advances in bulky and rigid linkers (such as cyclohexyl or benzylic motifs, e.g. Leng et al. *J Med Chem* 2025, doi: 10.1021/acs.jmedchem.4c01903) which extend from the N-substituent and occupy a hydrophobic pocket of VHL. This explains our rigid linker library, e.g. the introduction of 1,4-cubane as a suitable exit vector modification. We agree with this reviewer that in future studies for optimizing USP7-targeting PROTACs, an exit vector hopping strategy should be considered to further diversify the chemical matter. However, as our synthesized degraders NK250 and NK266 met all our criteria for potent and selective USP7 PROTACs, we decided to not develop these compounds further as part of this study.

We hope this additional information better explains the logic of our extended medicinal chemistry campaign, and that this reviewer supports our decision on the presented structures and SAR.

2. Would the authors be able to provide the WB data degradation data (Fig 2 c&d) quantification eg. analysed by densitometry and translated into a graphical representation so highlighting the SAR for the linker

We thank the reviewer for this comment and have added these data to Fig. 2c-d, as requested, in the form of a heatmap. We agree that this inclusion improves the clarity of the data. While degradation efficiencies differ between both cell lines, consistently, the precursors of our third generation PROTACs (NK225 and NK233) are the only molecules performing well in both cell lines.

3. Could the authors comment if they have considered or can characterizing the degrader ternary complex “cooperativity” via a biochemical or biophysical method to add to their characterization data of the degraders – this may provide data on efficiency of degradation and non-useful or optimal ternary complex formation of their molecules

We thank the reviewer for this suggestion as we also deem ternary complex formation cooperativity to give valuable insights into degrader mechanisms. Several biochemical and biophysical methods (such as SPR or ITC) were considered for the characterization of the ternary complex formation; however, we selected a cellular BRET-based assay as the most appropriate approach. This method enabled direct

monitoring of the interaction between USP7 and VHL within their physiological, cellular environment. We regard this to most accurately represent the natural context of degrader-induced ternary complex formation (see Fig. 4 and Supporting Fig. 4) and have therefore abstained from biochemical assays.

[editorial note: unpublished, confidential information redacted]

4. Could the authors comment further on the “degradation kinetics correlating with ternary complex formation” (P14In443) as this reviewer does not believe this data is at hand

We thank the reviewer for this comment. We have performed additional degradation kinetics assays (see new panels in Supporting Fig. 2e+f, complementing Fig. 3h+i) and added a clarification statement in the results section. Taken together, our data now support that only the fast degraders (e.g. NK250), but not the slow degraders (e.g. NK266), induce a stable ternary complex (see Fig. 4d). This is in agreement with literature demonstrating that stable ternary complexes favor rapid degradation kinetics (see Roy et al. *ACS Chem Biol* 2019, doi: 10.1021/acscchembio.9b00092).

For minor edits:

P6In168: provide reference for Buchwald-Hartwig

We have added a citation to the manuscript.

P9In278: missing word after central... mantra, tenant, paradigm ?

Thank you, we have added the word paradigm to improve the clarity of this sentence.

P10In312: is the resynthesis rate known for USP7?

We acknowledge that protein turnover rates (including synthesis and endogenous degradation) are crucial for targeted protein degradability. However, to our knowledge, the turnover rates of USP7 in these cell lines, including its resynthesis rate, have not been determined. Studies in different cell lines suggested USP7 to have a half-life of >24 h (Mathieson et al. *Nat Commun* 2018, doi: 10.1038/s41467-018-03106-1; Yi et al. *Cell Death Differ* 2023, doi: 10.1038/s41418-023-01180-7). This implies a rather low synthesis rate, in line with the pronounced and long-lasting USP7 degradation effect by our PROTACs. Indeed, we observed consistent depletion of USP7 beyond 72 h in wash-out experiments.

SI:

could the authors review C13 data for NK192, NK195 ... all analogs where they are reporting “d” peaks as this needs to be rectified

We thank the reviewer for highlighting this effect. We confirm the appearance of apparent doublets (termed “d”) in the ^{13}C NMR spectra of all compounds that comprise the 3-phenylbutyric acid moiety. Its chiral centre in combination with the hydroxypiperidine motif induce diastereomeric “rotamers” stemming from hindered rotation around the amide bond on the NMR-relevant time scale. This leads to apparent doublets in all ^{13}C NMR spectra which we annotate as “d” to separate them from “real” doublets, as rotamers are chemically distinct entities.

These rotamer peaks have previously been observed in spectra of similar compounds, e.g. see Gavory et al. *Nat Chem Biol* 2018, doi: 10.1038/nchembio.2528, there annotated as “conformers”, or Lamberto et al. *Cell Chem Biol* 2017, doi.org: 10.1016/j.chembiol.2017.09.003, see raw data enclosed).

To improve the visual and logical clarity, we have further highlighted the description of this effect by adding a figure at the start of the synthetic chemistry section of the Supporting Information (see page 12).

Reviewer #2 (Remarks to the Author):

The revised manuscript has been substantially improved and fully addresses all major and minor concerns raised in the initial review. The authors have responded thoroughly and convincingly, supported by extensive new experimental data that significantly strengthen the rigour and mechanistic depth of the study.

The previously noted dose inconsistency in the proteomics experiments has been comprehensively resolved through dose-matched analyses and supporting validation experiments, which confirm that the broader proteomic changes observed upon USP7 inhibition are modality-specific rather than concentration-driven. The rationale for prioritising DIA proteomics is now clear, and the inclusion of DDA data as supporting evidence is appropriate.

The authors have also added strong biological and functional validation, including pathway enrichment, transcriptomic analyses, and genetic USP7 knockout studies, which collectively provide compelling mechanistic insight into the consequences of USP7 degradation versus inhibition. The metabolic phenotyping is now clearly framed as an inhibitor-associated, USP7-independent effect, further strengthening the conceptual clarity of the work.

All minor issues related to data presentation and clarity have been addressed. The only remaining minor issue is that the label of one compound, which I think is "NK250", appears to be missing in Supporting Figure 7d and should be added.

Overall, the manuscript now represents a high-quality and well-substantiated contribution to the field. I recommend publication in its current form, pending this minor correction.

Thank you, we have added NK266 as compound label in Supporting Fig. 7d. We thank the reviewer for their careful evaluation of our manuscript and for the supportive recommendation.

Reviewer #3 (Remarks to the Author):

This reviewer commends the authors for their thorough and complete responses to the previous comments for the manuscript received from all of the reviewers. The inclusion of the additional data and clarification points and context of the positions taken in presenting their work have significantly improved the manuscript. As a result of the additional data and edits this reviewer would recommend this be published without further revision [editorial note: confidential, unpublished data redacted]

We thank the reviewer for their careful evaluation of our manuscript and for the supportive recommendation.

Review:

Targeted degradation of USP7 in solid cancer cells reveals disparate effects of deubiquitinase inhibition vs. acute protein depletion

Nikolas Klink; Sebastian Urban; Johanna A. Seier; Bikash Adhikari; Martin P. Schwalm; Juliane Muller⁵, Madeleine Dorsch; Farnusch Kaschani; Johannes Koch; Siska Fuhrer; Markus Kaiser; Nina Schulze; Stefan Knapp; Elmar Wolf; Annette Paschen; Barbara M. Gruner; Malte Gersch

Overall impression.

The article describes the synthesis USP7 VHL derived PROTAC array and thereafter characterization of the commensurate degradation of USP7 in 2 cell lines (melanoma and pancreatic ductal) by the compounds synthesized in this array. The manuscript also describes a BRET assay providing insight into ternary complex formation along with proteomic data associated with the treatment of those cells with an optimal ligand from the array. In contrast to these data, treatments with USP7 inhibitors uncovered contrasting effects at the cellular level including metabolic changes induced by prolonged (USP7 inhibitor) treatment. Overall, the authors claim to have provided a toolbox of comprehensive reagents resulting from this work, and more generally a strategy to distinguish the catalytic activities of DUBs from their non-catalytic functions.

The manuscript was well written.

This reviewer would support publication in Nature Communications, with minor revision following the consideration of the following points:

This reviewer would ask the authors

1. to strengthen the chemistry view of the VHL-degrader “library” and its design: specifically,
 1. Could they comment on and describe the design elements that drove the selection of the derivatives made in the “second generation” library: this reviewer would consider 15 compounds covering the chemical space is a good starting point but not comprehensive
 2. Were other vectors and chemistries considered, from the VHL ligand itself, beyond the N-substituent that they exemplified
2. Would the authors be able to provide the WB data degradation data (Fig 2 c&d) quantification eg. analysed by densitometry and translated into a graphical representation so highlighting the SAR for the linker
3. Could the authors comment if they have considered or can characterizing the degrader ternary complex “cooperativity” via a biochemical or biophysical method to add to their characterization data of the degraders – this may provide data on efficiency of degradation and non-useful or optimal ternary complex formation of their molecules
4. Could the authors comment further on the “degradation kinetics correlating with ternary complex formation” (P14In443) as this reviewer does not believe this data is at hand

For minor edits:

P6In168: provide reference for Buchwald-Hartwig

P9In278: missing word after central... mantra, tenant, paradigm ?

P10In312: is the resynthesis rate known for USP7?

SI:

could the authors review C13 data for NK192, NK195 ... all analogs where they are reporting "d" peaks as this needs to be rectified